# Bacterial lipid droplets bind to DNA via an intermediary protein that enhances survival under stress

Congyan Zhang[1,2], Li Yang[1], Yunfeng Ding[1], Yang Wang[1], Lan Lan[1,2], Qin Ma[2,3], Xiang Chi[1,2], Peng Wei[1,2], Yongfang Zhao[1,2], Alexander Steinbüchel[4,5], Hong Zhang[1,2] & Pingsheng Liu[1,2]

Lipid droplets (LDs) are multi-functional organelles consisting of a neutral lipid core surrounded by a phospholipid monolayer, and exist in organisms ranging from bacteria to humans. Here we study the functions of LDs in the oleaginous bacterium *Rhodococcus jostii*. We show that these LDs bind to genomic DNA through the major LD protein, MLDS, which increases survival rate of the bacterial cells under nutritional and genotoxic stress. MLDS expression is regulated by a transcriptional regulator, MLDSR, that binds to the operator and promoter of the operon encoding both proteins. LDs sequester MLDSR, controlling its availability for transcriptional regulation. Our findings support the idea that bacterial LDs can regulate nucleic acid function and facilitate bacterial survival under stress.

[1] National Laboratory of Biomacromolecules, CAS Center for Excellence in Biomacromolecules, Institute of Biophysics, Chinese Academy of Sciences, Beijing 100101, China. [2] University of Chinese Academy of Sciences, Beijing 100049, China. [3] Key Laboratory of Genomic and Precision Medicine, Beijing Institute of Genomics, Chinese Academy of Sciences, Beijing 100101, China. [4] Institute of Molecular Microbiology and Biotechnology, University of Münster, Corrensstrasse 3, D-48149 Münster, Germany. [5] Environmental Science Department, King Abdulaziz University, Jeddah 21589, Saudi Arabia. Correspondence and requests for materials should be addressed to P.L. (email: pliu@ibp.ac.cn).

The lipid droplet (LD) is a cellular organelle that has been found in almost all organisms[1–5]. LDs are involved in lipid storage and metabolism, intracellular molecular trafficking, and signalling[6–8]. LDs serve a basic set of functions associated with lipid homeostasis in all cells, but also have specialized functions in different cell types. As important sites of neutral lipid storage they are vital to human health[9], both as a source of nutrition as well as a site for energy storage. There is also significant interest in triacylglycerol (TAG) stored in bacterial LDs, especially in actinobacteria such as the genera *Mycobacterium*, *Nocardia*, *Rhodococcus*, *Micromonospora*, *Dietzia* and *Gordonia*, and in some streptomycetes[10]. Culture of these bacteria may serve as a starting material for the development of biodiesel. The diverse functions and extensive distribution of LDs, especially in many bacteria, suggest that LDs are essential organelles in cells and may have an ancient origin.

*Rhodococcus opacus* PD630 (PD630) and *R. jostii* RHA1 (RHA1) are oleaginous bacteria that contain large amounts of LDs and TAGs[11–13]. In an effort to gain a better understanding of LD functions and aid in developing a better biodiesel feedstock, we previously sequenced the genome and transcriptome of PD630, and analysed the LD proteome of PD630 and RHA1 (refs 14,15). We identified one major LD protein, microorganism lipid droplet small (MLDS) protein, that affects the size and content of LDs[15]. MLDS shares a conserved apolipoprotein motif with eukaryotic LD-resident proteins[4].

The features that distinguish the LD from other membrane-bound organelles are its neutral lipid core and its monolayer phospholipid membrane[16]. The specific chemical and physical properties of the membrane may contribute to the specificity and efficiency with which LDs participate in certain cell processes. Those physical features, along with the highly conserved nature of the LD proteome and the ability of LD proteins from evolutionarily distant organisms to be accurately targeted, suggest an ancient origin for the organelle[17,18]. As possibly one of the earliest evolved organelles, the LD may be vital for fundamental life processes including functions that have since been assumed by other organelles in eukaryotes.

There are several lines of evidence that indicate that LDs have a role in nucleic acid handling. A number of DNA- and RNA-related proteins were found in the LD proteomes of RHA1 and PD630 (refs 14,15). Furthermore, histones were found localized to LDs via the anchor protein Jabba in *Drosophila*[19], and LDs have been found in the nucleus of mammalian cells[20–22]. The nucleus serves as a major site of storage and regulation of nucleic acid function in eukaryotes, and we were curious whether the LD serves some related functions in bacteria.

Here we report that LDs in RHA1 bind to genomic DNA through MLDS, which increases the resistance of the bacterium and its genomic DNA to stress conditions. Furthermore, LDs control the expression of MLDS via the transcriptional regulator MLDS regulator (MLDSR). These data suggest that LDs may be involved in the regulation of nucleic acid function in bacteria.

## Results

**Lipid droplets bind genomic DNA through MLDS.** In our previous LD proteomic analyses of PD630 and RHA1, some putative DNA-related proteins were identified[14,15] (Supplementary Table 1), suggesting that LDs might be involved in nucleic acid function. Among these proteins, the major LD protein MLDS with N-terminal LD-targeting domain[15] contains four PAKKA motifs in its C-terminus (Fig. 1a, Supplementary Table 2). PAKKA motifs have been shown to bind DNA in prokaryotic cells, and are similar to the S/TPKKA motif of eukaryotic histone H1 (ref. 23). MLDS also contains

sequence similarities with Hlp (histone like protein) from mycobacteria and histone H1 from *Homo sapiens* (Supplementary Fig. 1a). Therefore, we wondered whether bacterial LDs might bind genomic DNA through their major protein, MLDS.

To determine if genomic DNA is associated with LDs, DNA and LDs were stained with SYTO9 and LipidTOX Red, respectively, in the wild-type (WT) RHA1, and were visualized by super-resolution structured illumination microscopy (SIM) (Fig. 1b). The images show that genomic DNA and LDs were associated (Fig. 1b). In contrast, deletion of MLDS markedly reduced the association between genomic DNA and LDs (Fig. 1b). Expressing a construct containing the LD-targeting domain of MLDS, but lacking the PAKKA motifs (MLDS[N]), did not rescue the mutant phenotype (Supplementary Fig. 1b). We then utilized a LD-associated protein, RHA1_ro05869 (ref. 15) as a LD marker protein to perform co-localization assays (Supplementary Fig. 1c,d). We stained DNA with SYTOX Orange and then quantified the ratio of genomic DNA co-localized with RHA1_ro05869-GFP relative to the total genomic DNA in the cells. We found a greater ratio of LD-associated genomic DNA in WT cells (~45%) than in MLDS knockout cells (~20%) (Fig. 1c,d). To obtain more accurate measurements, 48 genes were then selected randomly and were measured by polymerase chain reaction (PCR) using isolated LDs or other cell fractions from RHA1 as the templates (the purity of LD fraction was determined as shown in Supplementary Fig. 1f) (Fig. 1e, Supplementary Fig. 1e). More genes were found in the WT LD fraction (~80%) than in the MLDS knockout LD fraction (~60%) (Fig. 1f). Together, these data demonstrate that genomic DNA is partially localized on bacterial LDs and support that MLDS is a major LD-associated DNA-binding protein.

**Lipid droplets bind and recruit DNA via MLDS *in vitro*.** To gain insight into the mechanism by which MLDS binds DNA, a single-molecule pull-down assay was carried out (Fig. 2a). The signal from DNA and MLDS binding was about threefold higher than the signal from MLDS alone (Fig. 2b,c). Next, electrophoretic mobility shift assays (EMSAs) were conducted and showed that MLDS bound to DNA without DNA sequence specificity (Fig. 2d). Since the C-terminus of MLDS contains the positively charged lysine that is predicted to possess the ability to binding DNA (Fig. 2e), an EMSA was conducted with the WT C-terminal lysine rich region (GST-MLDS[C]) and a mutant C-terminal region with the lysine replaced by glutamic acid (GST-MLDS[C]). In contrast to the WT, the mutants lost the ability to bind DNA (Fig. 2f). The finding was then confirmed by surface plasmon resonance (SPR) (Supplementary Fig. 2a). In addition, EMSA also revealed that MLDS C-terminus deletion mutant (MLDS[N]) could not bind to DNA (Supplementary Fig. 2b). These results demonstrate that MLDS is able to bind DNA through its C-terminal PAKKA motifs.

To determine if MLDS recruits DNA to LDs, we constructed adiposomes, a spherical structure with a TAG core and a monolayer phospholipid membrane that can mimic cytosolic LDs[24] (Fig. 2g). DNA and proteins were then incubated with the adiposomes for 30 min at room temperature, and then the mixture was centrifuged to separate the adiposomes from the mixture solution (Fig. 2h). The results showed that both DNA and MLDS were recruited to the adiposome fraction (Fig. 2i, Supplementary Fig. 2c,d). It was also found that MLDS[N] was able to localize to adiposomes, but could not recruit DNA (Fig. 2i, Supplementary Fig. 2c,d), consistent with *in vivo* experiments (Supplementary Fig. 1b). In addition, we found that even though MLDS was also in the solution fraction, no DNA was detected in

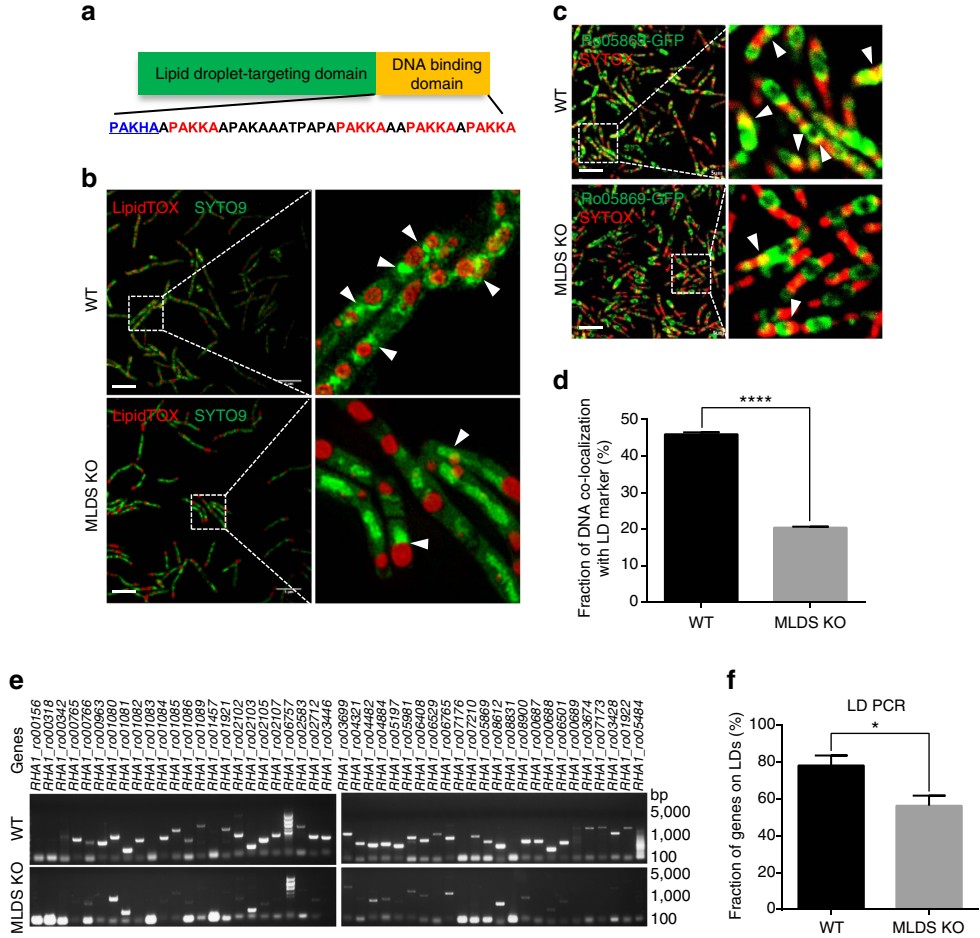

**Figure 1 | Lipid droplets bind genomic DNA via MLDS in RHA1 cells.** (**a**) Diagrams of MLDS domains and its C-terminal amino acid sequence. LD-targeting and putative DNA-binding domains and PAKKA motifs were indicated in green, orange, and red, respectively. (**b**) SIM images of wild type (WT) and MLDS knockout (MLDS KO) RHA1. LDs and DNA were stained with LipidTOX Red (red) and SYTO9 (green), respectively. Scale bar, 5 μm. The association of genomic DNA with LDs was indicated (white arrows). (**c**) Confocal microscopy images of WT and MLDS KO cells overexpressed RHA1_ro05869-GFP. The green 'ring-like' structure represents RHA1_ro05869-GFP targeting to LDs. DNA was stained by SYTOX (red). Scale bar, 5 μm. The co-localization of genomic DNA with RHA1_ro05869-GFP was indicated (white arrows). (**d**) Quantification of the fraction of DNA co-localized with RHA1_ro05869-GFP relative to the whole genomic DNA in **c**. Data represent mean ± s.e.m., $n = 100$ images. ****$P < 0.0001$, two-tailed $t$-test. (**e**,**f**) PCR analyses of LDs in RHA1. (**e**) 48 randomly selected genes were checked by PCR using LD fractions from WT and MLDS KO cells as templates. (**f**) Quantification of detected genes on LD fractions. Data represent mean ± s.e.m., $n = 4$. *$P < 0.05$, two-tailed $t$-test.

the solution fraction (Fig. 2i, Supplementary Fig. 2d), suggesting that LD-associated MLDS has a substantially higher DNA-binding affinity. The results indicate that, in RHA1 cells, LD-associated MLDS binds DNA with a much higher affinity than when in solution.

Next, we constructed two fusion proteins, MLDS[N]-H1 (MLDS N-terminus and histone H1) and ADRP-MLDS[C] (ADRP (adipose differentiation related protein, a LD-resident protein in mammalian cells[25]) and MLDS C-terminus) to perform adiposome-binding assays. Using these fusions we found that both proteins could bind DNA to the adiposomes (Supplementary Fig. 2e). The previous reports revealed that ADRP targets to adiposomes[24] and that histones localize to LDs via the anchor protein Jabba in *Drosophila*[26]. Our results (Supplementary Fig. 2e) indicate that the LD-targeting and DNA-binding domains of MLDS are at its N-terminus and C-terminus, respectively, which is consistent with the above results (Fig. 2f,i). They also support that LDs may be able to bind DNA through protein intermediates in eukaryotic cells. Together, the experiments described above indicated that MLDS binds DNA and recruits DNA to LDs.

In the RHA1 LD proteome, another LD-associated protein, RHA1_ro00689, contains seven PAKKA motifs (Supplementary Table 2). Based on its sequence similarity with the LD-associated histone anchor protein, Jabba, in *Drosophila* (Supplementary Fig. 3a), we named it Jabba-like protein (JLP). We found that JLP localizes to LDs by its N-terminal domain (Supplementary Fig. 3b,c), and that its C-terminus also binds DNA without sequence specificity (Supplementary Fig. 3d,e). However, in an adiposome-binding assay similar to the one described in Fig. 2h, no or little DNA signal was detected on adiposomes (Fig. 2j, Supplementary Fig. 3f), indicating that JLP does not recruit DNA to adiposomes. This finding suggests that there are at least two functional classes of DNA-binding proteins on LDs in RHA1. One type, such as MLDS, binds DNA either on LDs or in cytosol. The other one, represented by JLP, only binds DNA in the cytosol (Supplementary Fig. 3g–j).

**LDs binding to DNA contributes to survival under stress.** The experiments described above showed that genomic DNA is localized to LDs via MLDS in RHA1. Determining the

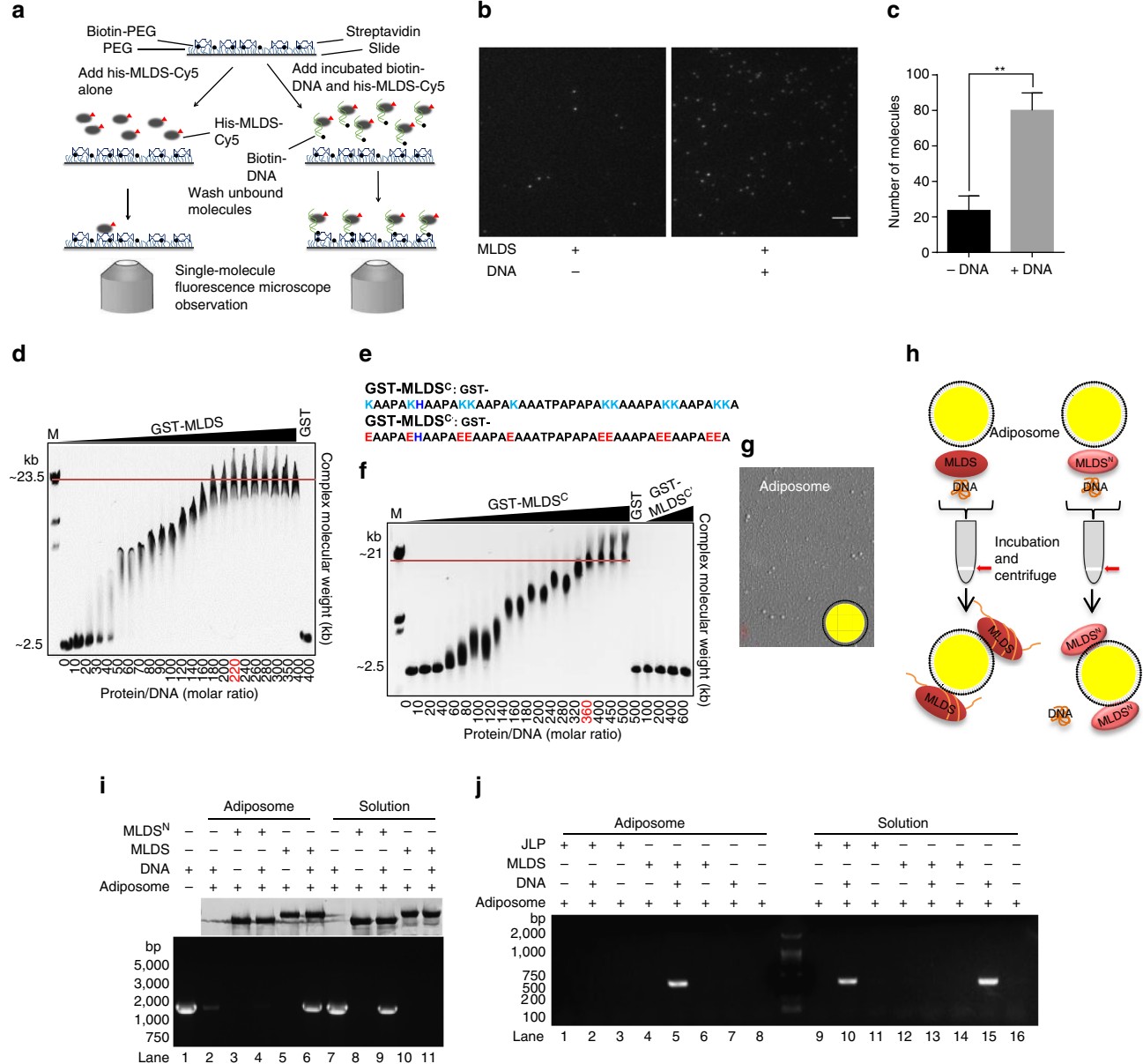

**Figure 2 | Lipid droplets bind DNA through MLDS *in vitro*.** (**a**–**c**) Interaction between DNA and MLDS was detected by single-molecule pull-down assay. Biotin-labelled DNA was incubated with Cy5-labelled MLDS, and the mixture was immobilized onto the streptavidin-coated coverslips and Cy5-labelled MLDS was observed. Sketch (**a**), validation (**b**) and quantification (**c**) of single-molecule pull-down assay were shown. The Cy5-labled MLDS non-specifically bound to the PEG-passivated surface is shown in left. Scale bar, 3.5 μm. Data represent mean ± s.d., $n = 3$. **$P < 0.01$, two-tailed *t*-test. (**d**) EMSA analysis of the interaction between DNA and MLDS. A dose dependent binding of GST-MLDS was obtained. Although these gel shifts were diffused when the protein/DNA ratio was from 20 to 40, these bindings could still be gradually increased since free monomer DNA was decreased with increased protein, and the number of GST-MLDS molecules bound to the 2.5 kb DNA could be saturated at 220 per DNA molecule (red). (**e**) The amino acid sequence of the MLDS C-terminus (MLDS$^C$) and the MLDS C-terminal mutant (MLDS$^{C'}$). All lysine in the region was replaced by glutamic acids. (**f**) EMSA analysis of MLDS$^C$ and MLDS$^{C'}$ with DNA. The 2.5 kb DNA molecule bound a maximum of 360 molecules of MLDS$^C$ (red). (**g**) The morphology and model of adiposomes. (**h**,**i**) Sketch (**h**) and validation (**i**) of adiposome-binding assay using adiposomes, DNA, and MLDS or MLDS N-terminus (MLDS$^N$). DNA, proteins and adiposomes were incubated and then the reaction was centrifuged to separate the adiposomes from the solution. DNA was detected by PCR and EB stained agarose gel (bottom). Protein was detected by silver staining (top). Lanes 4 and 9: MLDS$^N$, DNA and adiposomes; lanes 6 and 11: MLDS, DNA and adiposomes. Lanes 2–6 represented adiposome samples; lanes 7–11 represented solution samples. MLDS$^N$ and MLDS represented GST- MLDS$^N$ and GST-MLDS, respectively. (**j**) Similar to **i**, validation of adiposome-binding assay using adiposomes, DNA, and MLDS or JLP. Lanes 2 and 10: JLP, DNA and adiposomes; lanes 5 and 13: MLDS, DNA and adiposomes. JLP represented GST-JLP.

physiological function of this binding was our next goal. Previous studies have demonstrated that LD formation and TAG content are increased when bacteria are cultured under low nitrogen conditions[27]. Thus, we hypothesized that LDs might contribute to protection of the bacterium from certain stress conditions. To test

this idea, WT and MLDS knockout bacterial cells were cultured under extremely low nitrogen condition ($0.1 \, \mathrm{g \, l^{-1}}$ NH$_4$Cl). First, we examined the expression of MLDS in WT cells and found that the transcriptional level of MLDS was higher when the cells were cultured in the extremely low nitrogen condition, compared with

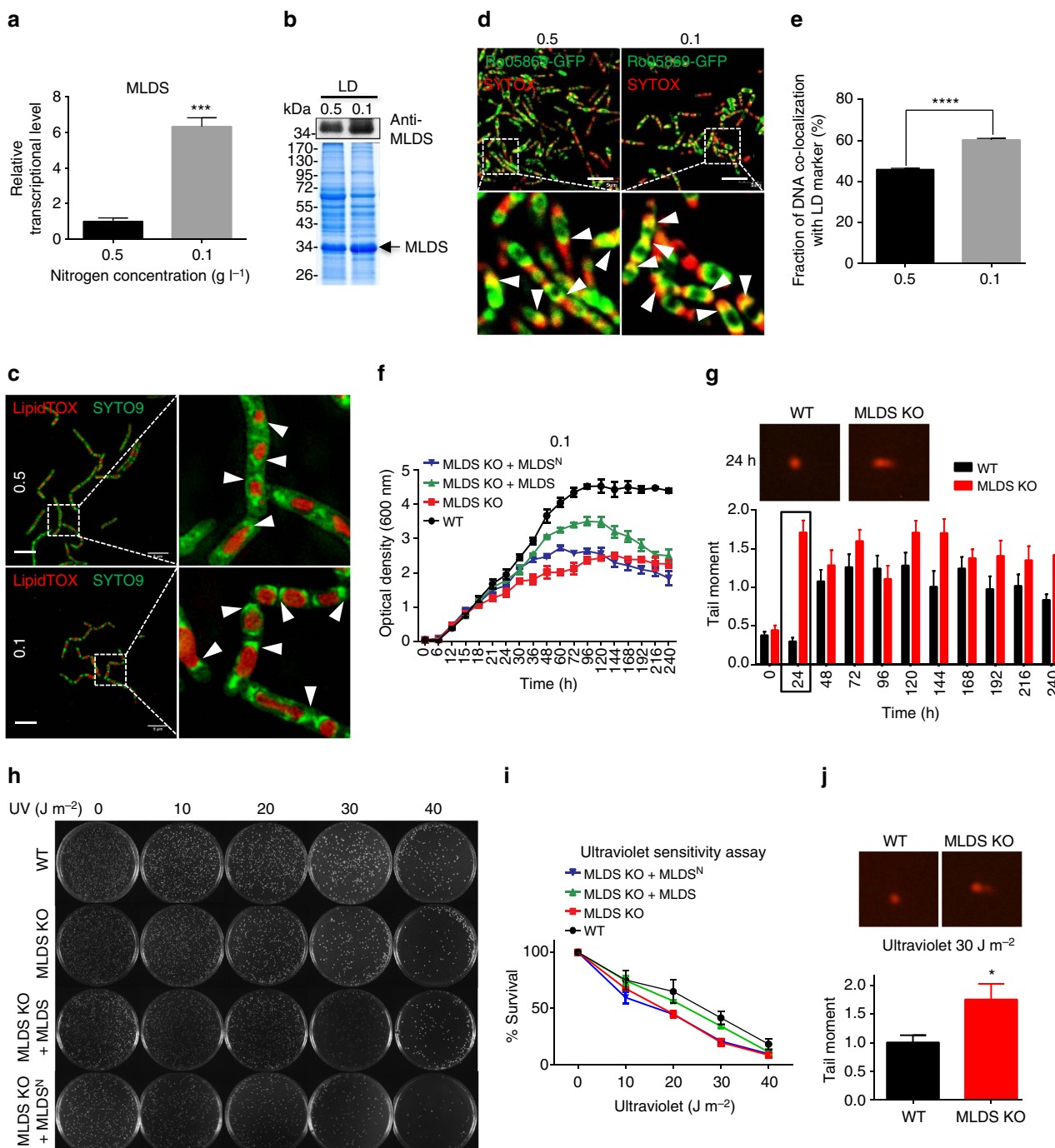

**Figure 3 | Lipid droplet-associated MLDS provides a survival advantage to RHA1 under stress.** (**a,b**) The transcriptional (**a**) and protein (**b**) levels of MLDS were measured in different nitrogen conditions. Data represent mean ± s.e.m., $n = 3$. ***$P < 0.001$, two-tailed $t$-test. (**c**) The SIM images of DNA (green) localized to LDs (red) in RHA1 which were cultured in MSM with 0.5 or 0.1 g l$^{-1}$ NH$_4$Cl. Scale bar, 5 μm. The association of genomic DNA with LDs was indicated (white arrows). (**d**) DNA (red) co-localized with RHA1_ro05869-GFP in RHA1 cultured in MSM with 0.5 or 0.1 g l$^{-1}$ NH$_4$Cl. Scale bar, 5 μm. The co-localization of genomic DNA with RHA1_ro05869-GFP was indicated (white arrows). (**e**) Quantification of the fraction of DNA co-localized with RHA1_ro05869-GFP relative to the whole genomic DNA in **d**. Data represent mean ± s.e.m., $n = 100$ images. ****$P < 0.0001$, two-tailed $t$-test. (**f**) The growth curve of control WT, MLDS KO, MLDS KO + MLDS and MLDS KO + MLDS$^N$ strains in MSM with 0.1 g l$^{-1}$ NH$_4$Cl. Data represent mean ± s.d., $n = 3$. (**g**) Neutral bacterial comet assays were performed on cells with or without MLDS cultured in extremely low nitrogen conditions every 24 h. The comet assay images and quantification of tail moment are shown in (top) and (bottom). DNA (red) in cells was stained with propidium iodide (PI). Data represent mean ± s.e.m., $n = 50$ cells. (**h,i**) Ultraviolet sensitivity assay. (**h**) WT and MLDS KO strains were treated with ultraviolet exposure (0, 10, 20, 30 and 40 J m$^{-2}$). (**i**) Quantification of survival rate. Data represent mean ± s.e.m., $n = 3$. (**j**) The neutral bacterial comet assay was performed on WT and MLDS KO cells following ultraviolet treatment (30 J m$^{-2}$). (Top) Comet assay images. (Bottom) Quantification of tail moment. Data represent mean ± s.e.m., $n = 50$ cells. *$P < 0.05$, two-tailed $t$-test.

the low nitrogen condition ($0.5\,g\,l^{-1}$ $NH_4Cl$) (Fig. 3a). Furthermore, a greater amount of MLDS was detected on isolated LDs from cells grown in the extremely low nitrogen condition (Fig. 3b). In addition to the induction of MLDS expression, the association of genomic DNA with LDs was increased with decreasing medium nitrogen as revealed in SIM images (Fig. 3c). Co-localization analysis also showed that more genomic DNA co-localized with LD marker in the extremely low nitrogen condition (Fig. 3d,e).

To determine if the responses described above are associated with a protective role under extreme conditions, we cultured WT and MLDS-knockout RHA1 cells in an extremely low nitrogen medium and monitored their growth. We found that the survival rate and growth rate (especially in late logarithmic phase) of WT cells in lower nitrogen were higher than MLDS KO cells (Fig. 3f, Supplementary Fig. 4a). Re-expression of MLDS, but not MLDS[N], partially rescued this phenotype (Fig. 3f, Supplementary Fig. 4a). Although the reason why MLDS could not fully recover the phenotype remains unknown, the result suggests that the DNA-binding region of MLDS is necessary to protect RHA1 cells from the effects of low nitrogen. To determine whether association of genomic DNA with LDs was able to protect the DNA from damage, we conducted the neutral bacterial comet assay[28] every 24 h on cells cultured in extremely low nitrogen conditions. This experiment showed that there were fewer DNA-strand breaks in the WT RHA1 than in the MLDS knockout mutant, with the greatest difference manifesting at 24 h (Fig. 3g, black frame).

The findings above were extended by examining the role of another genotoxic stressor, ultraviolet light. When RHA1 cells were exposed to ultraviolet radiation we found that WT cells had more surviving clones than MLDS knockout cells with or without overexpressed MLDS[N] (Fig. 3h,i), suggesting that the association of genomic DNA with LDs was able to protect the cells from ultraviolet damage. To verify this, cells were exposed to $30\,J\,m^{-2}$ ultraviolet treatment, and were then analysed with the neutral bacterial comet assay. The results revealed that a greater number of DNA-strand breaks occurred in cells without MLDS compared to WT cells (Fig. 3j). Furthermore, we found that LD association of a conserved DNA lesion-sensing endonuclease, UvrA (ref. 29) was reduced significantly in MLDS KO cells (Supplementary Fig. 4b–d), suggesting that LDs might be involved in the DNA repair process via the nucleotide excision repair system. We found that deletion of JLP had no effect on binding of genomic DNA to LDs or bacterial survival in extreme conditions (Supplementary Fig. 4e–j), further supporting that JLP is not responsible for recruiting DNA to LDs (Fig. 2j). Collectively, these data indicated that MLDS-mediated association of genomic DNA to LDs contributes to protection against certain stress conditions.

**MLDS expression is regulated by LD-associated MLDSR.**
We wanted to gain insight into how MLDS expression, and in turn LD binding of genomic DNA, are regulated in RHA1. When LD protein profiles of WT and MLDS KO were compared using SDS–polyacrylamide gel electrophoresis (PAGE), MLDS levels were greatly reduced while another band appeared in the MLDS KO mutant (Fig. 4a, red arrow). The mass spectrometry analysis showed that this band contained a putative transcriptional regulator, RHA1_ro02105. The N-terminus of this protein was predicted by the START database to contain a xenobiotic response element helix-turn-helix DNA-binding motif (Supplementary Fig. 5a). The *RHA1_ro02105* gene region overlaps slightly with the *mlds* gene in the genome and it is predicted that the two genes are in the same operon (Supplementary

Fig. 5a). Western blotting demonstrated that RHA1_ro02105 was a LD-associated protein (Fig. 4b). This was confirmed using confocal microscopy and western blot analysis of GFP-fused RHA1_ro02105 in WT and MLDS KO cells (Fig. 4c,d). To elucidate its function, the gene was then either overexpressed or knocked out in RHA1 without changing the *cis*-element or coding region of *mlds* (Supplementary Fig. 5b–e). We found that the mRNA and protein levels of MLDS were significantly reduced in both the deletion and overexpression mutants (Fig. 4e,f), suggesting that the protein may possess both positive and negative transcriptional regulation activity. Hence, RHA1_ro02105 was termed MLDSR.

To dissect the mechanism by which MLDSR regulates MLDS expression, we searched for the DNA motif that MLDSR binds. The 103 bp sequence making up the upstream region of the *mldsr* and *mlds* operon is illustrated in Fig. 5a. EMSA analysis demonstrated that MLDSR could bind the 103 bp sequence as well as a shorter 43 bp sequence with two mobility shift bands (Fig. 5a,b, Supplementary Fig. 6a–f), indicating that there could be two MLDSR-binding motifs in the 43 bp. We further narrowed down the sequence to illuminate the two putative motifs and identified them in the 43 bp DNA (motif (1 + 2)) including a 20 bp (motif 1) and a 23 bp (motif 2) regions (Fig. 5c). The interactions between MLDSR and motifs 1 and 2 were confirmed using super-EMSA analysis (Supplementary Fig. 6g). The binding affinities were determined using SPR ($K_{D(motif\ 1+2)} = 62.0\,nM$, $K_{D(motif\ 1)} = 175.0\,nM$ and $K_{D(motif\ 2)} = 5.14\,\mu M$) (Fig. 5d–f, Supplementary Fig. 6h). Furthermore, we performed constitutive mutation assays to ascertain the precise DNA-binding site of MLDSR (Supplementary Fig. 6i,j, red) and define a 16 bp palindromic sequence, 5′-GNT (T/A) GCTNNTGCTANC-3′, as the MLDSR-binding box, which functions as a specific binding sequence for MLDSR (Fig. 5g, bottom). In addition, it was found that these motifs were partially symmetrical (Fig. 5g, middle). We also found that MLDSR did not bind single stranded DNA, and the directivity and space distance of DNA were necessary for MLDSR binding (Supplementary Fig. 6k,l).

Based on the affinity and similarity, we speculated that motif 1 and motif 2 were the operator and promoter of the operon, respectively. To test this, we first performed a promoter expression assay by transforming plasmids with or without motif 2 (Fig. 5h) into RHA1 cells. The results showed that motif 2 could drive red fluorescence protein (RFP) and his-tag expression in RHA1 (Fig. 5i,j), suggesting that motif 2 was the promoter. We then carried out an *in vitro* transcription assay using three different DNA templates (Fig. 5k, Supplementary Fig. 6m). It was shown that MLDSR could repress transcription by binding motif 1 (Fig. 5l) (protein/DNA ratio > 50:1) and that the repression occurred at transcription initiation (Supplementary Fig. 6n). Furthermore, we found that MLDSR could regulate transcription positively when the protein/DNA ratio was low (2:1–6:1) (Fig. 5m). The results demonstrated that motif 1 is the operator region and indicated that MLDSR may regulate transcription by affecting RNA polymerase binding to the promoter. These results are consistent with the finding that both MLDSR KO and MLDSR overexpression reduced MLDS expression in RHA1 (Fig. 4e,f). In summary, MLDSR can bind specifically to two motifs (promoter and operator) of the operon and regulate transcription of MLDS and MLDSR positively and negatively.

**LDs control MLDSR function by regulating its effective levels.**
Since MLDSR is associated with LDs, it seems possible that LDs are involved in the regulation of MLDSR activity. To test this possibility, we first identified the LD targeting and DNA-binding regions of MLDSR using truncation mutations. Bioinformatics

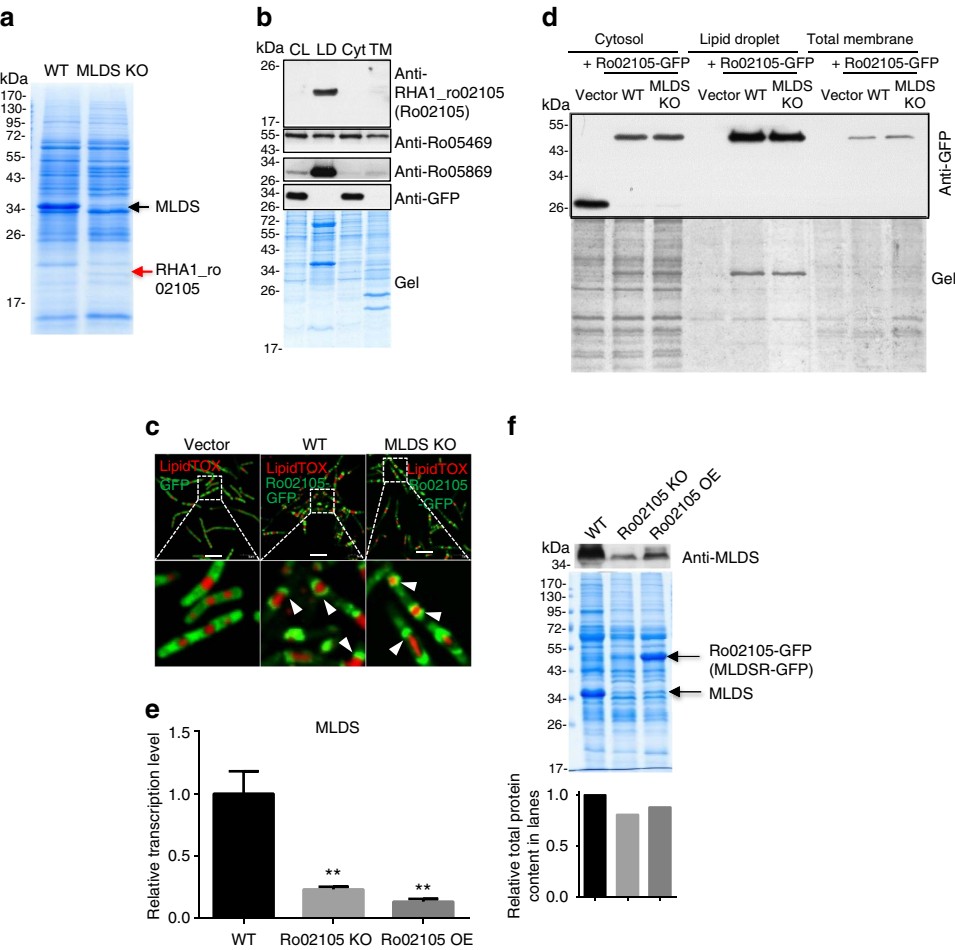

**Figure 4 | Expression of MLDS is regulated by transcriptional regulator MLDSR. (a)** RHA1_ro02105 was identified on LDs in MLDS KO RHA1 by mass spectrometry (MS) analysis (red arrow). Gel was stained with Colloidal Blue and the band was cut for MS identification. **(b)** Western blot analysis of cell fractions of RHA1 WT + GFP cells using anti-Ro02105, anti-Ro05469, anti-Ro05869 and anti-GFP. CL, whole cell lysate; LD, lipid droplet; Cyt, cytosol; TM, total membrane. Gel was stained with Colloidal Blue. **(c)** Confocal microscopy images of RHA1_ro02105-GFP locating on LDs in WT and MLDS KO cells. The LD targeting of RHA1-ro02105-GFP was indicated (white arrows). LDs were stained with LipidTOX Red (red). Scale bar, 5 μm. **(d)** RHA1_ro02105-GFP located on LDs in WT and MLDS KO cells by western blot analysis of cell fractions of vector, WT and MLDS KO cells. Gel was stained with Colloidal Blue. **(e)** The transcriptional level of MLDS in RHA1_ro02105 KO and OE cells was measured. Data represent mean ± s.e.m., n = 3. **P < 0.01, two-way ANOVA. **(f)** The protein level of MLDS in RHA1_ro02105 KO and OE cells was measured. The arrows indicate MLDS and MLDSR-GFP. The bar graph at bottom represents the relative total protein content in the lanes through quantification using Image J software.

analysis and circular dichroism data predicted that MLDSR contains seven α-helices (Supplementary Fig. 7a,b). The locations of GFP-fused truncations, made based on the predicted α-helices, were determined (expression level shown in Supplementary Fig. 7c). Deletion of either α2 or α6 caused MLDSR to lose LD localization and locate to the cytosol, while full-length MLDSR-GFP was on LDs, implicating that helices α2 and α6 are involved in LD targeting (Fig. 6a,b). Furthermore, an EMSA with truncated proteins revealed that the proteins without α2 could not bind DNA (Fig. 6c,d). In addition, since MLDSR overexpression repressed MLDS transcription in RHA1 (Fig. 4e), we over-expressed these truncated proteins in RHA1 and found that α2–α4 repressed MLDS transcription (Fig. 6e). Thus, the LD localization and DNA-binding regions of MLDSR share at least one α-helix, α2 (Supplementary Fig. 7k), which suggests there might be competitive binding of MLDSR to LDs and DNA.

Next, we performed an *in vitro* assay designed similarly to that described above. DNA, MLDSR and adiposomes were incubated by three manners as shown in Fig. 6f, and then the reaction was centrifuged to separate the adiposome and solution fractions (Fig. 6f). The result revealed that although most MLDSR could be

recruited to adiposomes, the DNA was not detected in the same fraction (Fig. 6g, lanes 4–6), suggesting that adiposomes and DNA compete for MLDSR binding and thus LD-localized MLDSR loses its DNA-binding ability. To further verify whether LD localization of MLDSR affects its transcriptional regulatory function, we utilized the solution samples of the adiposome-binding assay similar to Fig. 6g as templates to perform *in vitro* transcription assays. With the addition of increasing quantities of adiposomes, MLDSR was increased in the adiposome fraction and decreased in the solution (Fig. 6h, top), and the relative transcriptional level was enhanced even higher than control without MLDSR (Fig. 6h, middle and bottom), suggesting that only cytosolic MLDSR could regulate transcription. This result is consistent with the observation that MLDSR regulates transcription both positively and negatively (Fig. 5l,m). Altogether, these results support that at low concentrations, MLDSR may positively regulate expression of both MLDS and itself (Fig. 5m). When MLDSR concentration becomes high, MLDSR may repress the expression of both MLDS and itself (Fig. 5l). If cells need more MLDS, LDs may recruit MLDSR to reduce the concentration of MLDSR in cytosol, thus driving MLDS expression (Fig. 6h,i).

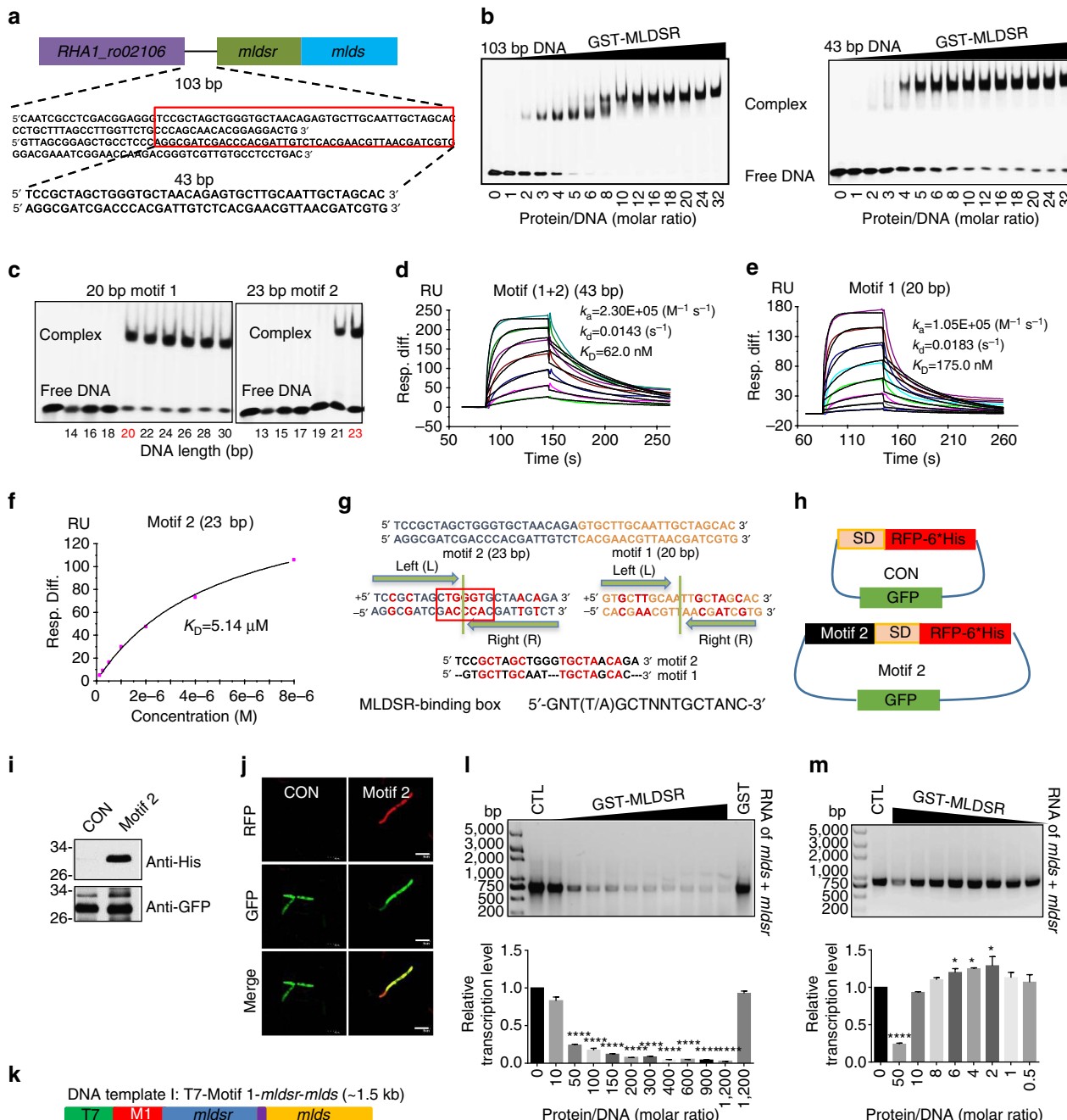

**Figure 5 | MLDSR regulates MLDS expression by binding the promoter and operator of their operon.** (**a**) Schematic showing DNA motif for MLDSR binding and that *mlds* and *mldsr* are in the same operon. (**b**) EMSAs of MLDSR binding the 103 bp and 43 bp DNA motif. (**c**) EMSAs of MLDSR binding 20 bp (motif 1) and 23 bp (motif 2). (**d–f**) SPR analysis for MLDSR binding 43 bp (motif (1+2)) (**d**) 20 bp (motif 1) (**e**) and 23 bp (motif 2) (**f**). Kinetic analysis and steady state affinity were used in the assays, respectively. (**g**) (top) Schematic showing 43 bp DNA consisting of motif 2 and motif 1. (middle) Motif 2 and motif 1 are palindromic sequences. The key base pairs in the two motifs are shown in red. Motif 2 is similar with T7, T3 and SP6 promoters (shown in red frame). (bottom) The palindromic MLDSR-binding box is shown. (**h**) Sketch of two plasmids in promoter expression assay. (**i**) Western blotting of CON and Motif 2 strains using anti-His and anti-GFP. (**j**) Confocal microscopy images of control and motif 2 strains. Scale bar, 5 μm. (**k–m**) *In vitro* transcription assays. (**k**) Sketch of DNA template. The *in vitro* transcription assay using DNA template I under high protein/DNA ratio (>50:1) (**l**) or low protein/DNA ratio (<10:1) (**m**). CTL represents no MLDSR in reaction and GST represents GST, but not GST-MLDSR in reaction. RNA was detected by EB stained agarose gel (top). Quantification was performed using Image J software (bottom). Data represent mean ± s.e.m., n = 3. *P < 0.05, ***P < 0.001, ****P < 0.0001, two-way ANOVA.

The symmetrical property of motif 1 and motif 2 (Fig. 5g) suggests that the MLDSR protein might form a symmetrical oligomer, similarly to the λ repressor[30]. To test this, a co-immunoprecipitation experiment was carried out and showed that MLDSR could interact with itself *in vivo* (Supplementary Fig. 7d). Experiments using the crosslinker glutaraldehyde revealed that MLDSR from isolated LDs and purified GST-MLDSR, but not GST alone could form oligomers (Supplementary Fig. 7e–g). Furthermore, we found that the α5–α6 domain was required for MLDSR oligomerization

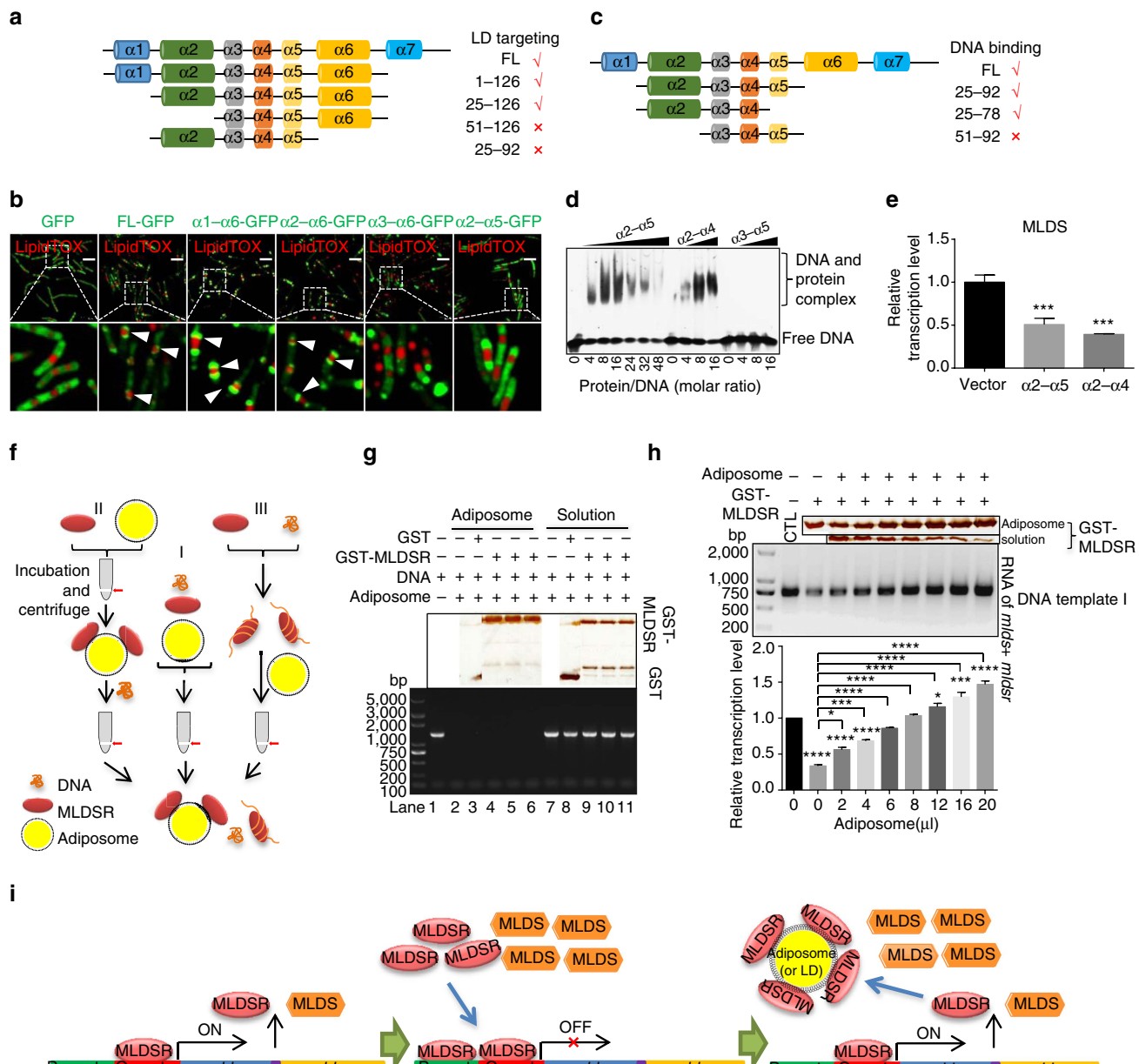

**Figure 6 | Lipid droplets mediate MLDSR transcriptional regulation by controlling its effective cytosolic concentration. (a)** Sketch of the full-length MLDSR and truncation mutants (1–126, 25–126, 51–126 and 25–92 amino acids). (**b**) Confocal microscopy images of localization of different truncation mutants fused with GFP in bacteria. The LD targeting of truncation mutants was indicated (white arrows). The location of α3–α6-GFP and α2–α5-GFP is similar to free GFP location. LDs were stained with LipidTOX Red (red). Scale bar, 5 μm. (**c–e**) The truncated proteins (25–92, 25–78 and 51–92 amino acids) (**c**) were purified for EMSA (**d**) and overexpressed in RHA1 for qRT-PCR analysis (**e**). Data represent mean ± s.e.m., n = 3. ***P < 0.001, two-way ANOVA. (**f,g**) Sketch (**f**) and validation (**g**) of adiposome-binding assay using adiposomes, DNA (DNA template I), and MLDSR by three manners: I, MLDSR, DNA and adiposome incubated together, lanes 4 and 9; II, MLDSR and adiposomes incubated firstly, and then DNA was added, lanes 5 and 10; III, MLDSR and DNA incubated firstly, and then adiposomes were added, lanes 6 and 11. DNA was detected by PCR and EB stained agarose gel (bottom). Protein was detected by silver staining (top). Lanes 2–6 represented adiposome samples; lanes 7–11 represented solution samples. (**h**) In vitro transcription assays using the solution fractions of adiposome-binding assay by manner I in **g** as templates. Proteins in adiposome and solution fractions were detected by silver staining (top). RNA was detected by EB stained agarose gel (middle). Quantification was performed using by Image J software (bottom). CTL represents no MLDSR in reaction. Data represent mean ± s.e.m., n = 3. *P < 0.05, ***P < 0.001, ****P < 0.0001, two-way ANOVA. (**i**) Schematic diagram of MLDSR regulating expression of both MLDS and itself when MLDSR concentration is low (left) or high (middle) and adiposomes are added (right).

(Supplementary Fig. 7h–j). Together, these findings suggest that MLDSR can locate on LDs and bind DNA as oligomers (Supplementary Fig. 7k–o).

**MLDSR contributes to survival under stress.** We next examined the effect of MLDSR on LD binding to genomic DNA and

protection of the cells from extreme conditions. First, we examined the association between LDs and genomic DNA in MLDSR knockout (MLDSR KO), MLDSR overexpressed (MLDSR OE) and WT cells. Association of genomic DNA and LDs was reduced markedly in MLDSR KO and OE cells compared with WT cells (Fig. 7a). This is consistent with the

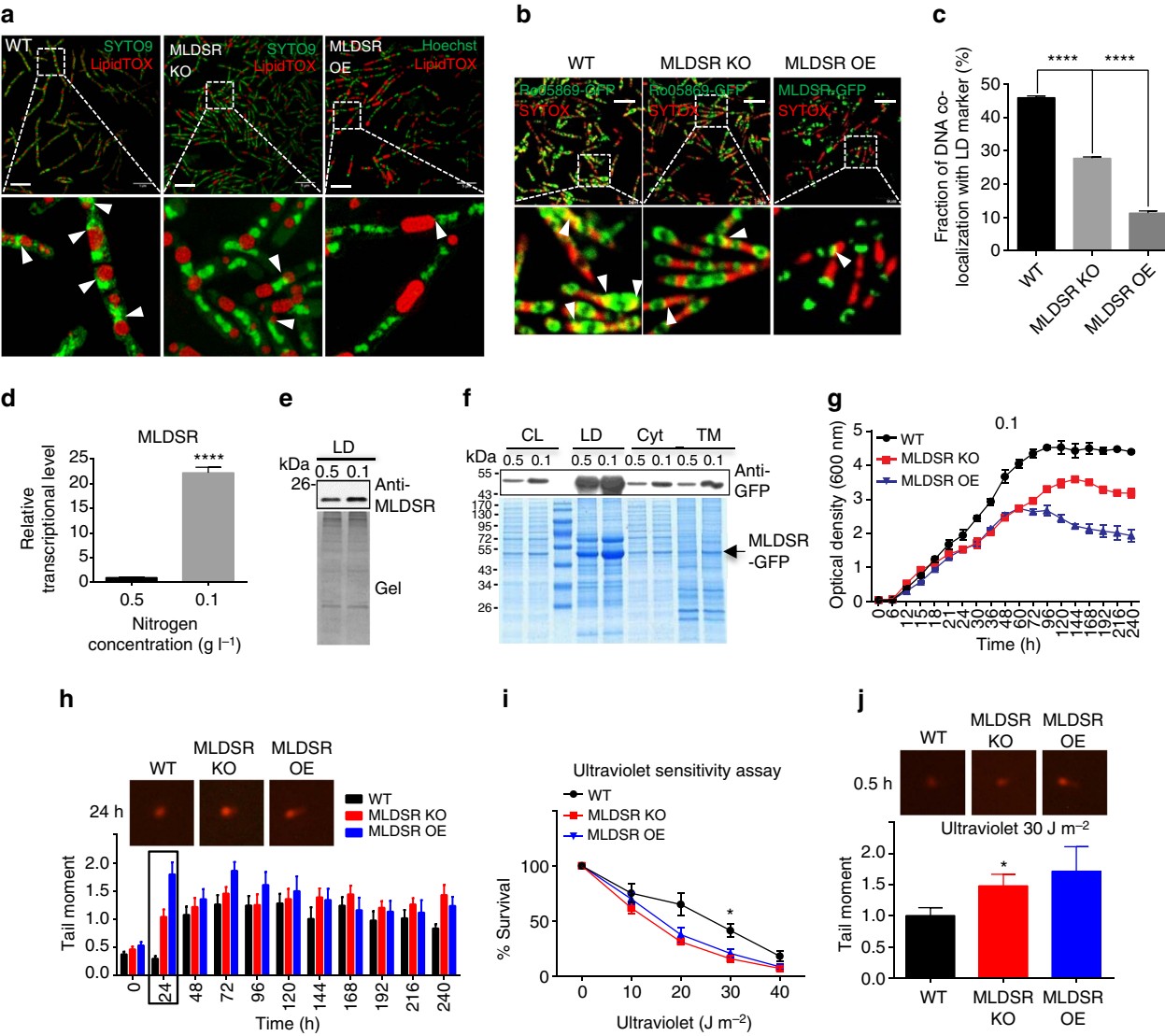

**Figure 7 | MLDSR affects RHA1 survival by regulating MLDS expression.** (**a**) The SIM images of DNA (green) locating on LDs (red) in WT and MLDSR KO, MLDSR OE strains. Scale bar, 5 μm. The association of genomic DNA with LDs was indicated (white arrows). (**b**) The confocal microscopy images of DNA (red) co-localization with LD marker protein in three strains. Scale bar, 5 μm. The co-localization of genomic DNA with RHA1_ro05869-GFP was indicated (white arrows). (**c**) Quantification of the fraction of DNA co-localized with RHA1_ro05869-GFP relative to the whole genomic DNA in **b**. Data represent mean ± s.e.m., n = 100 images. ****P < 0.0001, two-way ANOVA. (**d,e**) The transcriptional (**d**) and protein (**e**) levels of MLDSR in RHA1 cultured in different mediums were measured. Data represent mean ± s.e.m., n = 3. ****P < 0.0001, two-tailed t-test. (**f**) The distribution of MLDSR-GFP in RHA1 cultivated in different mediums was measured. The arrow indicates MLDSR-GFP. (**g**) The growth curve of WT, MLDSR KO and MLDSR OE strains in MSM with 0.1 g l$^{-1}$ NH$_4$Cl. Data represent mean ± s.d., n = 3. (**h**) Neutral bacterial comet assays for WT, MLDSR KO and MLDSR OE cells cultivated in extremely low nitrogen conditions every 24 h. The comet assay images and quantification of tail moment were shown at top and bottom respectively. Data represent mean ± s.e.m., n = 50 cells. (**i**) Ultraviolet sensitivity assay. WT, MLDSR KO and MLDSR OE strains were treated with ultraviolet exposure (0, 10, 20, 30 and 40 J m$^{-2}$). Data represent mean ± s.e.m., n = 3. *P < 0.05, two-way ANOVA. (**j**) Neutral bacterial comet assays were performed following ultraviolet treatment (30 J m$^{-2}$) on WT, MLDSR KO and MLDSR OE cells. Top: comet assay images. Bottom: quantification of tail moment. Data represent mean ± s.e.m., n = 50 cells. *P < 0.05, two-way ANOVA.

results that deletion of MLDS decreased the binding of genomic DNA to LDs (Fig. 1b) and both deletion and overexpression of MLDSR reduced MLDS expression (Fig. 4e,f). Furthermore, co-localization assays also showed a decreased ratio of genomic DNA co-localization with a LD marker related to total genomic DNA in MLDSR knockout (∼27%) and overexpressed cells (∼15%) compared with WT cells (Fig. 7b,c).

Second, we detected the expression of MLDSR when cells were cultured in different nitrogen conditions. It was found that when cells were cultured in extremely low nitrogen conditions, transcription of *mldsr* was induced (Fig. 7d) and LD-associated

MLDSR was increased (Fig. 7e). Furthermore, it was confirmed that more MLDSR-GFP located on LDs when cells were cultivated under extremely low nitrogen conditions (Fig. 7f). The results suggest that the expression of MLDS could be stimulated continually at a lower nitrogen condition (Fig. 3a,b) because the co-expressed MLDSR was bound by LDs to maintain the MLDSR concentration in cytosol in a positive regulatory range, which is consistent with *in vitro* experiments (Fig. 6h,i).

Finally, since MLDSR regulates MLDS expression, we determined if MLDSR is involved in the protection of RHA1 under stress. The same experiments used to examine the role of

MLDS under stress conditions were conducted and results showed that, similar to the MLDS deletion mutant, the survival and growth rates (especially in late logarithmic phase) of both MLDSR KO and MLDSR OE strains were lower (Fig. 7g), and DNA-strand breaks were higher for both strains (Fig. 7h) in extremely low nitrogen condition than WT cells. In addition, both MLDSR KO and OE cells were more sensitive to ultraviolet irradiation (Fig. 7i) and had more DNA-strand breaks after ultraviolet treatment than WT cells (Fig. 7j). Furthermore, it was shown that DNA lesion-sensing endonuclease UvrA partially lost its LD localization in MLDSR KO cells (Supplementary Fig. 8a–c). Overall, our results indicate that the expression level and localization of MLDSR regulates MLDS expression. This in turn modulates MLDS-mediated binding of genomic DNA to LDs, which appears to contribute to DNA stabilization and enhanced survival under certain stress conditions (Fig. 8).

## Discussion

The LD is a multi-functional cellular organelle found in many organisms including eukaryotes and some bacteria[1–13]. In a previous study, we identified in the *Caenorhabditis elegans* LD proteome the most abundant LD protein for that organism, MDT-28 (ref. 31). Its mammalian homologue is a mediator of RNA polymerase II, mediator complex subunit 28 (MED-28)[32].

Previous LD proteomic analyses also have identified RNA-binding proteins, ribosomal subunits and translation factors on LDs[15,31,33]. Ribosomes[34] and RNA[35,36] are localized to LDs in leucocytes and mast cells, respectively. Furthermore, histones H2A, H2B (refs 19,37) and H2Av (ref. 26) are located on LDs via the Jabba protein in *Drosophila*, and LD-associated histones are involved in antibacterial activity[38]. Hepatitis C virus is located at and assembles around the LD surface in mammalian cells[39–41]. LDs have also been found in the nucleus in eukaryotic cells[20–22]. In addition, the protein FSP27 on LDs inhibits the translocation of NFAT5 from the cytoplasm to the nucleus, thereby reducing NFAT5 transcriptional activity[42]. Collectively, these studies suggest that eukaryotic LDs can bind and regulate nucleic acid functions.

LDs are coated by a phospholipid monolayer membrane, and phospholipids are negatively charged or electrically neutral under physiological conditions. Thus, it seems unlikely that the polyanionic DNA chain may electrostatically interact with phospholipid membrane without intermediary proteins or other factors. Previously reported examples of proteins mediating interaction between DNA and the bacterial cell membrane include the protein Noc (ref. 43).

The previous finding of DNA-related proteins in the proteomes of actinobacterial LDs[14,15] raised the possibility that LDs might be

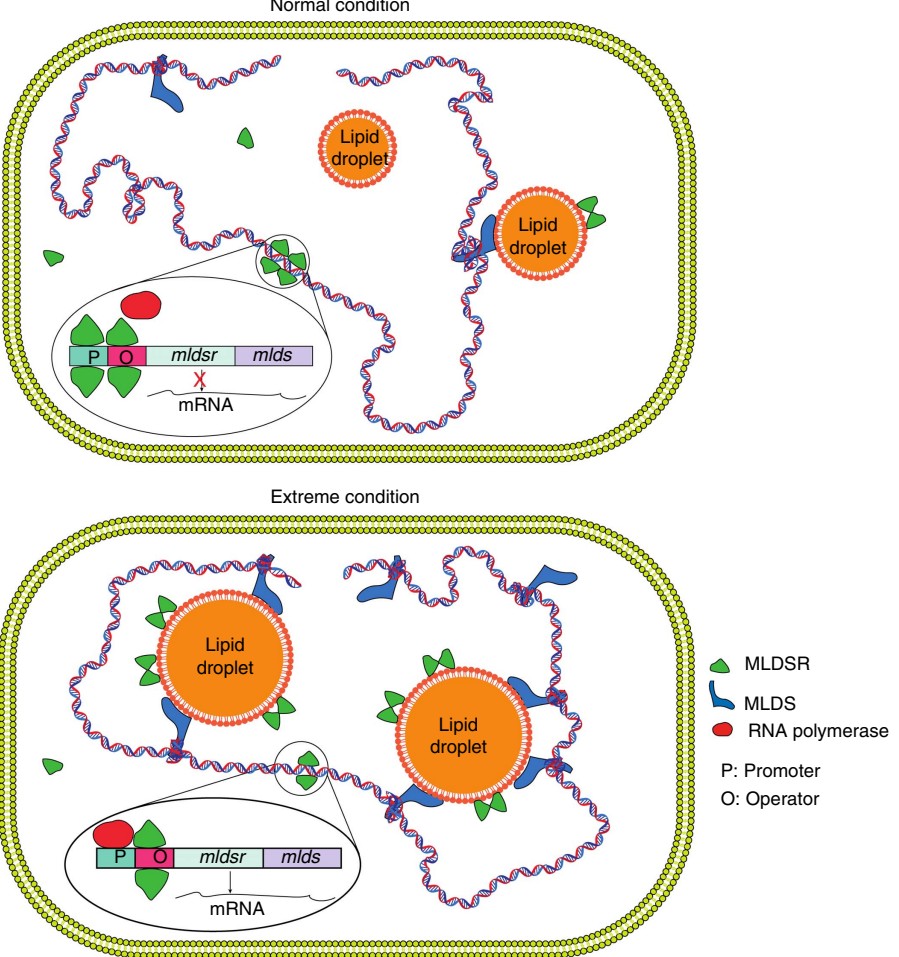

**Figure 8 | Proposed functions for LDs as well as MLDS and MLDSR during acclimation to stressful conditions.** Under normal conditions, MLDSR in the cytosol represses expression of both MLDSR and MLDS, which might be due to inhibition of RNA polymerase binding to the promoter. Since MLDS is a major protein that binds DNA to LDs, decreasing MLDS on LDs reduces binding of genomic DNA to the organelle. In contrast, under stressful conditions, more MLDSR molecules are translocated to LDs, reducing the cytosolic concentration, which in turn enhances the expression of MLDS and MLDSR. The increased MLDS on LDs drives the binding of genomic DNA to LDs, which exerts a protective effect, such as reducing DNA damage.

involved in DNA-related processes in these bacteria. Through cell-based and *in vitro* assays, we determined here that LDs bind and seem to stabilize genomic DNA through the major LD protein MLDS. We also found that bacterial LDs modulate MLDS expression via the transcriptional regulator MLDSR. In turn, MLDS provides a survival advantage to bacteria under certain stress conditions, apparently via its interaction with genomic DNA. Putative MLDS and MLDSR homologues are found in other actinobacteria (Supplementary Fig. 8d,e), suggesting that these LD functions might be common in this bacterial phylum.

In addition to MLDS, JLP and MLDSR, other DNA-related proteins have been identified in the LD proteomes of RHA1 and PD630 (Supplementary Table 1), including three proteins containing the PAKKA motif (Supplementary Table 2). Their role in binding genomic DNA to LDs is unclear, and we will study them in future research.

## Methods

**Bacterial strains and plasmids.** All *R. jostii* RHA1 and *Escherichia coli* strains and plasmids used in this study are listed in Supplementary Data 1. *E. coli* TOP10 was used for plasmid propagation and BL21 (DE3) for protein expression. Plasmid pJAM2 was used for protein expression in RHA1, pK18mobsacB for gene knockout in RHA1, and pGEX-6 P-2 and pET-28a for protein expression in *E. coli.*

**Cell cultivation.** All RHA1 cells were cultivated at 30 °C in LB medium to $OD_{600} \sim 2.0$, then 0.5 ml cells were transferred into 10 ml of mineral salt medium (MSM) with $0.5\,g\,l^{-1}$ (low nitrogen condition) or $0.1\,g\,l^{-1}$ (extremely low nitrogen condition) $NH_4Cl$ as the nitrogen source and were grown to $OD_{600} \sim 2.0$. In the study, MSM with $0.5\,g\,l^{-1}$ $NH_4Cl$ is used as normal condition and without any indications, RHA1 cells were transferred into MSM with $0.5\,g\,l^{-1}$ $NH_4Cl$ for experiments.

**Protein expression and purification.** The *mlds, mlds N-terminus, mlds C-terminus, mlds C-terminal mutant, jlp, jlp$_{2 + 3}$, mldsr* and three truncated *mldsr* (25–92, 25–78, and 51–92) genes were amplified from RHA1 genomic DNA (NC_008268) (primers shown in Supplementary Data 2) and were cloned into the pGEX-6 P-2 plasmid (GE Healthcare). These plasmids were transformed into *E. coli* BL21 (DE3). The proteins were expressed with an N-terminal GST tag. The cells were grown at 37 °C in LB to $OD_{600} \sim 0.6$, and were then induced for 3 h with 0.2 mM isopropyl-D-thiogalactopyranoside. The cells were collected and resuspended in lysis buffer 1 (20 mM Tris-HCl, pH 8.0, 0.5 M NaCl, 1 mM EDTA, 4% (v/v) glycerol, 1 mM DTT). The bacteria were broken and centrifuged at 39,191 *g* for 45 min. The proteins were purified with Glutathione Sepharose 4B (GE Healthcare) according to the manufacturer's instructions. For expression of MLDS with a His tag for use in single-molecule pull-down assay, the *mlds* gene was cloned into the pET-28a plasmid (Novagen). The plasmid was transformed into *E. coli* BL21 (DE3), and the growth and induction were conducted as described above. The cells were then collected and resuspended in lysis buffer 2 (20 mM Tris-HCl, pH 8.0, 0.5 M NaCl). After lysis and centrifugation, the proteins were purified with $Ni^{2+}$ Sepharose 6 Fast Flow (GE Healthcare) according to the manufacturer's instructions. The concentration of these purified proteins was determined by Nanodrop (Eppendorf).

**Construction of GFP-fused proteins in RHA1.** The *mlds, mlds N-terminus, mlds C-terminus, jlp*, the truncated *jlp, mldsr*, the truncated *mldsr* genes and other genes were amplified from RHA1 genomic DNA (primers shown in Supplementary Data 2) and were cloned into the pJAM2 plasmid containing the *gfp* gene with BamHI or ScaI site (the *gfp* gene was inserted into the pJAM2 with XbaI site firstly and GFP was at the C-terminus of these fusion proteins). These cloned plasmids were transformed into RHA1 cells using a Bio-Rad 165-2100 MicroPulser (Bio-Rad, USA). Positive clones were selected on LB agar plates containing $50\,\mu g\,ml^{-1}$ kanamycin and were screened using fluorescent microscopy.

**Sample preparation for confocal microscopy and SIM.** Cultivated RHA1 cells were added on cover glasses pretreated with poly-L-lysine (PB0589) for 30 min before washing. Cells were then incubated in a 1:500 solution of LipidTOX Red (H34476), SYTO9 (S34854), SYTOX Orange (S11368) or Hoechst in darkness for 30 min. Samples were mounted on glass slides using mounting media (P0126) and observed with an Olympus FV1000 confocal microscope or super-resolution SIM.

**Super-resolution 3D-SIM imaging.** 3D-SIM images were acquired on the DeltaVision OMX V3 imaging system (Applied Precision, GE) with a 100 × 1.512 oil objective (Olympus UPlanSApo), solid-state multimode lasers (488, 405 and 561 nm) and electron-multiplying charge-coupled device cameras (Evolve

512 × 512, Photometrics). Serial *z*-stack sectioning was done at 125 nm intervals. The microscope was routinely calibrated with 100 nm fluorescent spheres to calculate both the lateral and axial limits of image resolution. Acquisition settings were as follows: for LipidTOX Red, 10–30 ms exposure with 561 nm laser (100% gain); for SYTO9, 10–20 ms exposure with 488 nm laser (100% gain); Hoechst, 30–50 ms exposure with 405 nm laser (80% gain). The powers of the lasers are 500 mW (561 nm), 500 mW (488 nm), and 600 mW (405 nm), respectively. The %*T* numbers are 1.0–10.0 for 561 and 488 nm lasers and 31.3 for 405 nm laser. SIM image stacks were reconstructed using softWoRx 5.0 (Applied Precision) with the following settings: pixel size 39.5 nm; channel-specific optical transfer functions; Wiener filter constant 0.0020; discard Negative Intensities background; drift correction with respect to first angle; custom K0 guess angles for camera positions. Pixel registration was corrected to be <1 pixel for all channels using 100 nm Tetraspeck beads. These experiments were performed in triplicate.

For Hoechst staining of DNA, we replaced the original blue colour with green colour in images by softWoRx 5.0, without any other changes, to observe the association between the green of DNA and the red of LDs.

**Single-molecule pull-down.** Two complementary 15-nucleotide DNA oligomers were synthesized by TAKARA (Supplementary Data 2). One strand bore a 3′-biotin and 5′-NH2. The chemically synthesized DNA was resuspended in nuclease-free water and was diluted into 50 mM potassium borate buffer, pH 8.1, 200 mM KCl (ref. 44). The dye-labelled DNA was generated by adding more than ten-fold molar excess of Cy3-NHS (lumiprobe) to the non-biotinylated DNA and mixing in the dark for 2 h at 37 °C. The unbound dye was removed using a desalting column. The complementary strands were mixed in equimolar ratios, were heated to 75 °C, and were then cooled to room temperature passively to allow hybridization[44]. The concentration of DNA labelled by Cy3 was adjusted to 10 μM, which was used for testing the binding between DNA and the microscope slides. The unlabelled DNA was prepared using the same protocol to obtain hybridized DNA, and was adjusted to 200 nM. This preparation was used for detecting binding between DNA and His-MLDS protein. The experiment was carried out in triplicate.

**Single-molecule imaging.** Fluorescence experiments were performed using an objective based total internal reflection fluorescent microscope. Cy3 fluorophores was excited with a 532 nm laser (Coherent Inc., Sapphire SF). His-MLDS protein labelled with Cy5 was excited with 640 nm laser (Coherent Inc., Sapphire SF). Photons emitted from fluorescent dyes were collected using a 1.49 NA × 100 objective (Olympus UAPON × 100 OTIRF) and were detected with a cooled EMCCD (Andor iXon Ultra). Fluorescence data were acquired using the software Metamorph (Universal Imaging Corporation). For His-MLDS protein labelled with Cy5, the Em gain was set to 200, the exposure time was 50 ms, and the intensity was 15 mW. The coverslip was cleaned and coated successively with polyethylene glycol (PEG)/PEG-biotin and 100 μM streptavidin with a 2 min incubation time[45–47].

**Adiposome-binding assay.** Adiposomes were prepared using methods previously described[24]. In brief, 2 mg dried 1,2-di(9Z-octadecenoyl)-sn-glycero-3-phosphocholine and 5 μl rat TAG were vortexed with 100 μl buffer B (20 mM HEPES, 100 mM KCl, 2 mM MgCl₂, pH 7.4) for 24 pulses of 10 s. Following centrifugation the cloudy upper band and pellet were discarded and the remaining adiposome containing fraction was collected. The adiposomes, 2.5 kb DNA and proteins (GST-MLDS, GST-MLDS-RFP, $GST-MLDS^N$-RFP, GST-MLDSR, GST-JLP or GST) were incubated together or individually at room temperature for 30 min. The preparations were then centrifuged at 21,130 *g* for 5 min. The adiposomes were then washed three times with buffer B. The samples were adjusted to equal volumes. One aliquot was used as a template for PCR to detect DNA, and the other aliquot was dissolved in 2 × SDS loading buffer and denatured at 95 °C for 5 min. The protein samples were resolved by SDS–PAGE and silver stained. These experiments were repeated at least three times.

**Electrophoretic mobility shift assay.** The interaction between increasing amounts of GST-MLDS or $GST-MLDS^C$ or $GST-MLDS^N$ and 0.4 pmol of ~2.5 kb linear DNA (the molar ratio of protein to DNA was from 10:1 to 500:1) was carried out in binding buffer (20 mM Tris-HCl, pH 7.5, 150 mM NaCl, 5 mM MgCl₂, 1 mM EDTA, 1 mM DTT and 4% glycerol). The reactions were incubated at room temperature for 30–40 min and were then mixed with 6 × loading buffer and analysed by electrophoresis on a 0.5% agarose gel in 0.5 × TBE (890 mM Tris-boric acid and 20 mM EDTA, pH 8.3) at 30 V for 13–15 h. The gels were visualized by staining with ethidium bromide (EB). The binding of MLDSR proteins (full length, 25–92, 25–78 or 51–92 amino acids) and a variety of dsDNA probes (Supplementary Data 2) was characterized by native PAGE. DNA probes were from PCR products (>60 bp) or from synthesized and annealed oligonucleotides (<60 bp) including mutated DNA motif probes. The annealing buffer contained 50 mM HEPES pH 7.4 and 100 mM NaCl. Increasing amounts of protein were incubated with 400–500 ng of DNA at 24 °C for 30–40 min in a total of 15 μl binding buffer. The molar ratio of protein to DNA was from 1:1 to 32:1. The protein–DNA complex was mixed with 6 × loading buffer and then resolved on a 3.5% PAGE gel in 0.5 × TBE at 120 V for ~40 min. The gels were visualized by

staining with EB for 15 min. The quantification of complex/free DNA was performed with Image J software. These experiments were repeated at least three times.

**In vitro transcription assay.** To determine if motif 1 was the operator, the T7 promoter was placed upstream of motif 1 and the DNA sequence of genes *mldsr* and *mlds* (about 1,400 bp in total) and the linear DNA was used as a template for *in vitro* transcription assays. Increasing amounts of MLDSR protein was incubated with 0.3 pmol T7-motif 1 DNA in a total of 10 µl at room temperature for 30–40 min. The molar ratio of protein to DNA was from 0.5:1 to 1,200:1. Then the reaction was used as a template and RNA was synthesized according to the manufacturer's instructions (RiboMAX Large Scale RNA Production System-T7 P1300). The samples were resolved on a 1% agarose gel and visualized by EB staining. To confirm that the regulation occurred at transcriptional initiation, 20 bp or 400 bp of DNA were inserted between the T7 promoter and motif 1, which were used as templates. To verify that only cytosolic MLDSR regulated transcription, adiposomes, MLDSR and DNA were incubated together for 0.5 h, and then the reaction was centrifuged to separate the adiposomes. The solution component was used as a template for an *in vitro* transcription assay. The quantification of synthetic RNA was performed using Image J software. These experiments were carried out at least three times.

**Quantification of fluorescent co-localization.** Fluorescence intensity from SYTOX Orange staining and overexpressed GFP-tagged LD markers was quantified with Image J software. The LD markers used were RHA1_ro05869-GFP in WT, MLDS KO and MLDSR KO cells, and MLDSR-GFP in MLDSR-overexpressed cells. Co-localization was evaluated using the JACoP (ref. 48) plugin for Image J, by calculating Mander's fraction of the A image overlapping the B image, which represented the fraction of DNA co-localizing with a LD marker. In WT cells cultured in MSM with 0.5 g l$^{-1}$ NH$_4$Cl for control, it was used the mean threshold of 1,440 for SYTOX signal and the mean threshold of 670 for RHA1_ro05869-GFP signal. In MLDS KO cells, it was used the mean threshold of 1,330 for SYTOX signal and the mean threshold of 610 for RHA1_ro05869-GFP signal. In MLDSR KO cells, it was used the mean threshold of 1,150 for SYTOX signal and the mean threshold of 800 for RHA1_ro05869-GFP signal. In WT cells cultured in MSM with 0.1 g l$^{-1}$ NH$_4$Cl, it was used the mean threshold of 1,400 for SYTOX signal and the mean threshold of 760 for RHA1_ro05869-GFP signal. In MLDSR OE cells, it was used the mean threshold of 1,730 for SYTOX signal and the mean threshold of 1,150 for MLDSR-GFP signal. Fluorescence overlap was quantitatively assessed in 100 2D-confocal microscopy images with the middle z stack. A fractional value of 1 was defined as 100%. The colocalization analysis was modified from the previous method[49]. The experiment was carried out in triplicate.

**Bioinformatics analyses.** A PAKKA motif search was performed with BLAST in the NCBI database. Sequence similarity alignments of MLDS with histone H1 and Hlp, and JLP with Jabba were carried out by ClustalX2 software[50]. 3D structure models of MLDS, JLP and MLDSR were predicted using I-TASSER[51]. Secondary structure of MLDSR was predicted using the network protein sequence analysis (https://npsa-prabi.ibcp.fr) using the GOR4 method[52] and Phyre2 (ref. 53). The xenobiotic response element-helix-turn-helix domain of MLDSR was predicted using the START database.

**Ultraviolet sensitivity assay.** All RHA1 cells (WT, MLDS KO, MLDS KO + MLDS, MLDS KO + MLDS$^N$, MLDSR KO and MLDSR-overexpressed cells) were cultivated in LB medium to OD$_{600}$ ∼2.0 under late exponential phase, and were then diluted and spread on LB solid medium plates (the initial number of cells is ∼1,000). After drying the plates were exposed to ultraviolet light (0, 10, 20, 30 and 40 J m$^{-2}$) and were then cultivated at 30 °C for 3 days. The surviving clones were counted and were expressed as per cent surviving relative to the unexposed plates. These experiments were performed three times.

**Neutral bacterial comet assay.** Cells were pre-treated for the neutral bacterial comet assay. Cells that were exposed to ultraviolet were cultured for 0.5 h on solid media after exposure (20 J m$^{-2}$) and then clones were selected randomly and incubated in 100 µl of 20 mg ml$^{-1}$ lysozyme solution at 37 °C for 2 h to dissolve the cell wall. Cells that were cultured in extremely low nitrogen condition were incubated in lysozyme solution at 37 °C for 2 h to achieve protoplasts.

The neutral bacterial comet assay was carried out by combining previous methods[28,54,55]. In brief, two agarose layers were prepared. The second layer contained 2 µl of pre-treated RHA1 cells (WT, MLDS KO, MLDSR KO and MLDSR-overexpressed cells). The cells were lysed in lysis buffer (10 mM Tris-HCl, pH 10, 2.5 M NaCl, 100 mM EDTA, 1% sodium lauroyl sarcosine, 1% Triton X-100 and 10% DMSO) at room temperature for 2 h. Following lysis, the slides were immersed in 1 × TBE buffer at room temperature for 2 h to unwind the supercoiled DNA. Then the slides were electrophoresed in 1 × TBE buffer at 15 V and 15 mA for 50 min. After electrophoresis, the slides were dried and stained with 50 µl of propidium iodide for 5 min. The treated slides were observed with a Nikon eclipse Ti-U microscopy at × 40 magnification. The cell comets were analysed using CASP software and tail moment was measured statistically in 50 cells.

**Quantitative real-time-PCR analysis.** The experiments were performed using previous method[14]. In brief, RHA1 was cultured in MSM to OD$_{600}$ ∼2.0, and then total RNA was isolated using Trizol Reagent (Invitrogen) and purified using the TIANGEN RNAclean Kit (TIANGEN) according to the manufacturer's instructions. For qRT-PCR analysis, RNA was reverse transcribed using the M-MLV Reverse Transcriptase Kit (Promega) which was used in the qPCR reactions containing SYBR green fluorescent dye (ABI). The relative expression of mRNA was determined after normalization against 16S levels using the DD-Ct method, comparing MLDS or MLDSR expression level. qPCR was performed using an ABI StepOne PLUS PCR instrument. All primers used for qRT-PCR are shown in Supplementary Data 2. These experiments were performed in triplicate.

**Lipid droplet isolation.** LD isolation was performed using method previously described[56]. In brief, RHA1 cells were collected and resuspended in buffer A (25 mM tricine, 250 mM sucrose, pH 7.8). The resuspended cells were homogenized by passing through a French pressure cell three times at 100 MPa, 4 °C. The samples were then centrifuged at 6,000 g for 10 min. The supernatant fraction (10 ml) was overlaid with 2 ml of buffer B (20 mM HEPES, 100 mM KCl, 2 mM MgCl$_2$, pH 7.4) and was centrifuged at 38,000 r.p.m. for 1 h at 4 °C (Beckman SW40). The LDs were collected and washed three times with 200 µl of buffer B. To prepare the LD protein sample, 1 ml of chloroform:acetone (1:1, v/v) was added. LD proteins were extracted, dissolved in 2 × SDS sample buffer, and denatured at 95 °C for 5 min. For LD PCR experiments, isolated LDs were diluted in the same volume as whole cell lysate and 48 random genes (Supplementary Data 2) were detected by PCR. These experiments were conducted four times.

**Construction of JLP and MLDSR deletion mutants.** Deletion mutants of JLP and MLDSR were constructed by homologous recombination using previous method[15]. In brief, first, the upstream and downstream sequences of target gene were cloned by PCR using primers a/b and c/d to generate fragments AB and CD. And then the two fragments were ligated and inserted into the pK18mobsacB plasmid with EcoRI and HindIII sites. The cloned plasmids were transformed into RHA1 cells using a Bio-Rad 165-2100 MicroPulser (Bio-Rad, USA). Positive clones were selected on LB agar plates containing 50 µg ml$^{-1}$ kanamycin followed by sacB counter selection. Final confirmation of the mutant in kanamycin-sensitive, sucrose-resistant colonies was obtained by PCR using primers a and d. To further confirm that gene the target gene had been deleted, primers f and r were used to amplify this gene by PCR. All primers used are shown in Supplementary Data 2. Sequencing and western blot analyses were used to verify the absence of these genes and proteins.

**Cross-linking of MLDSR by glutaraldehyde.** Cross-linking of MLDSR was carried out using purified GST-MLDSR protein, purified LDs or RHA1 cells. First, GST-MLDSR was diluted in cross-linking buffer (20 mM Tris-HCl, pH 7.5, 150 mM NaCl, 5 mM MgCl$_2$, 1 mM EDTA, 1 mM DTT and 4% glycerol) to 310 nM and was incubated with glutaraldehyde at a final concentration of 2, 5 or 7 mM for 1 h at 24 °C. Second, purified LDs were incubated with glutaraldehyde for 1 h at room temperature. Third, RHA1 cells overexpressing MLDSR or its truncated proteins were broken in cross-linking buffer by sonication and were incubated with glutaraldehyde. The samples were resolved by SDS–PAGE and probed by western blot using anti-MLDSR antibody or anti-GFP antibodies.

**Antibody preparation.** The antibody of MLDSR (anti-MLDSR) was produced in the study. Two rabbits were immunized with a mixture of two synthetic peptides (Peptide 1: CEPEPPTVEQEKADD, Peptide 2: CESRTSELRTEEHQRSD). After three injections, the rabbit sera were tested using western blotting. The anti-MLDS, anti-Ro05469, and anti-Ro05869 were provided from the previous study[15]. These antibodies were used under 1:5,000 dilution. The anti-GFP (IMA1006L) and anti-His (IMA1005L) were purchased from the IMAGEN company. Several original scans of the most important western blots presented in the main figures (Figs 3b,4b,f and 7e) can be found in Supplementary Fig. 9.

**Promoter expression assay.** To confirm that motif 2 was a promoter, motif 2, the SD sequence, the *rfp* gene with 6 × His and a terminator sequence were inserted into pJAM2-GFP plasmid at the Acc65 I site (Motif 2 plasmid). The reconstructed plasmid was transformed into RHA1. A plasmid containing only the SD sequence, the *rfp* gene with 6 × His, and a terminator sequence was used as a control (CON plasmid). The *gfp* gene was regulated by a constitutive promoter, and the *rfp* gene with 6 × His was regulated by motif 2. GFP and RFP with his-tag expression were observed by confocal microscopy and were detected by western blotting with anti-GFP antibody and anti-His antibody (both were 1:2,000 dilution).

**Surface plasmon resonance.** To study the interaction between MLDSR and DNA, three pairs of complimentary primers (motif 1, motif 2 and motif 1 + 2) were synthesized. The forward primer was 5′-biotinylated. Pairs of primers were incubated to anneal. The products were used as DNA probes for SPR analysis.

SPR experiments were performed on a BIAcore 3000 biosensor system (Biacore AB, Sweden). The three biotinylated-DNA probes were covalently coupled to the surface of three different channels of an SA sensorchip according to the manufacturer's instructions, using the biotin as an anchor. The degree of capture was about 100 RU. Increasing concentrations (15.625–8,000 nM) of MLDSR were injected over the surface of the sensorchip at a flow rate of $30\,\mu l\,min^{-1}$ in SPR buffer (20 mM Tris-HCl, pH 7.5, 150 mM NaCl, 5 mM $MgCl_2$, 1 mM DTT and 0.005% Tween 20 (v/v)). The reaction time was 1 min and separation time was 2 min. After the reaction, 2 M NaCl was injected at a flow rate of $30\,\mu l\,min^{-1}$ for 90 s. The resulting sensorgrams were analysed using BIAevaluation 4.1 software according to steady state affinity. The 1:1 binding model was used to determine the kinetic constants. The $\chi^2$ values and the random distribution of the residuals were used to assess goodness of fit. These experiments were made in technical triplicates.

**Co-immunoprecipitation of MLDSR-GFP with anti-GFP antibody.** RHA1 cells were grown at 30 °C in MSM to $OD_{600} \sim 2.0$. The cells were collected and the cell wall was removed with lysozyme (20 mg ml$^{-1}$) at 37 °C for 2 h. The protoplasts were centrifuged and incubated with IP buffer TESTN150 (20 mM Tris-HCl, pH 8.0, 1 mM EDTA, 0.5% SDS, 1% Triton X-100, 150 mM NaCl) for 30 min. After centrifugation, the supernatant was incubated with the anti-GFP antibody (ab290) (the final concentration was 0.03 mg ml$^{-1}$) and rotated at 4 °C for 2 h. Protein A beads were added to the mixture and the reaction was rotated at 4 °C for 2 h. The reaction was then centrifuged and the supernatant and beads were both collected. The beads were washed three times with TESTN150 and the proteins were eluted with SDS loading buffer. The samples were denatured at 95 °C for 5 min and analysed by SDS–PAGE. The experiment was carried out in triplicate.

**Phylogenetic analyses.** For estimation of phylogenetic trees of the orthologous proteins (Supplementary Fig. 8), 20 taxa proteome sequences were retrieved from NCBI database. To find the orthologous proteins of MLDS and MLDSR, blastp were performed with parameter of –e value 1e − 5. MLDSR was discovered in 20 taxa and MLDS was discovered in 12 taxa. Then phylogenetic analyses of the orthologous proteins were done using MEGA v. 5 (ref. 57) for Maximum Likelihood analyses.

**Statistics.** Trial experiments or experiments done previously were used to determine sample size with adequate statistical power. No data were excluded in the statistical analysis of the study. The investigators who carried out the experiments were not completely blinded, but the data were randomly analysed in a blinded manner. The results are represented as a mean ± s.e.m. or s.d. of at least three independent experiments. Two-tailed Student's $t$-test or two-way ANOVA were used to evaluate the statistical difference of the results. Statistical significance was estimated when $P < 0.05$.

**Data availability.** The authors declare that all data supporting the findings of the study are available within the paper and its Supplementary Information files, or from the authors upon request.

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

## Acknowledgements

We thank Drs John Zehmer, Xuejun C. Zhang, Bao-Liang Song, Bin Liang and Yong Liu for their critical reading and useful suggestions. We thank Qinglong You for suggestion of protein purification, Shuoguo Li for taking and analysing SIM images, Yuanyuan Chen and Zhenwei Yang for SPR assay technical assistance and data analysis, and Lili An and Fanlei Ran for neutral bacterial comet assay technical assistance. This work was supported by grants from the Ministry of Science and Technology of China (Grant No. 2016YFA0500100), from the National Natural Science Foundation of China (Nos U1402225, 31571388, 61273228, 81270932 and 31301106) and from Chinese Academy of Sciences (Grant No. XDA12030200).

## Author contributions

P.L. conceived the project. C.Z. and P.L. designed the experiments. C.Z. performed the main experiments, and C.Z. and P.L. analysed the data. L.Y. and Y.D. found and identified JLP locating on LDs. Y.W. provided technique for generation of adiposomes. L.L. and Y.Z. performed and analysed single-molecular pull-down and image experiments. Q.M. conducted the phylogenetic analyses. X.C. contributed to ultraviolet sensitivity assay and cell growth experiments. P.W. designed and drew the proposed model pictures. Experiments and manuscript were assisted by contributions from A.S. and H.Z. Manuscript was written by C.Z. and P.L.

## Additional information

**Competing interests:** The authors declare no competing financial interests.

