## [Peer Review File · Nature Communications]

Reviewers' comments:

Reviewer #1 (Remarks to the Author):

This manuscript reports that the lipid droplet in *Rhodococcus jostii* binds and protects genomic DNA. This binding requires the MLDS protein. It is also reported that expression of MLDS depends on the regulator MLDSR.

The role of the lipid droplet in organizing and protecting genomic DNA is interesting, and the contribution of MLDS to DNA binding is convincing.

The proposed mechanism by which MLDSR controls MLDS is not quite as clear (particularly the activation), as detailed below (items 10-12):

1. Perhaps it is the reproduction, but I cannot see any labeling in Fig. 1f.
2. Please define abbreviations on first use (e.g., ADRP, line 126; DOPC, line 448).
3. Line 107. In Fig. 2f, does MLDS-C' correspond to just the lysine-rich region with the lysines replaced? Please clarify. Was full-length MLDS in which lysines were mutated tested? Or the N-terminal domain of MLDS? As it stands, this assay indicates that the lysine-rich C-terminus is involved in DNA binding, but it does not rule out that the N-terminal domain contributes.
4. Line 125/ Fig. 2i/ Fig. S2d. Some DNA is associated with adiposomes in the MLDS-N mutant as well as in ADRP. Would that suggest that DNA binding may occur either non-specifically or through the MLDS N-terminal domain?
5. Line 139. It is said that no DNA signal is seen with adiposomes and JLP, however, a weak signal is seen in Fig. S3f, lane 1? More careful wording may be required.
6. Line 168/ Fig. 3f. Why is growth phenotype only partially rescued by expression of MLDS?
7. Fig. 3g. Can the authors comment on why the comet assay only shows a difference after 24 h? No difference is seen after 48 h or longer.
8. Fig. 3h-i. Why was MLDS KO strain complemented with full-length MLDS not included?
9. Line 198. What was the justification for selecting this particular band? Please mention in the main text how this protein was identified. Is the operon predicted or confirmed? "orchestrated" may not be the best term here.
10. The binding assays for MLDSR (Fig. 5) are confusing. According to Methods, a single site binding model was used for determination of K_d from SPR assays, however, the DNA in Fig. 5d contains two sites, so this is not the correct binding model. The text (line 219) indicates a 5.14 μM K_d for motif 2, but Fig. 5f says millim. According to Methods, EMSA was performed with 400-500 ng DNA in a 15 μL reaction, corresponding to ~ 0.5 -1 μM DNA (depending on length). If the K_d for 43 bp DNA is 62 nM (Fig. 5d), that means EMSAs were performed under stoichiometric conditions; if this is so, then why does it require a stoichiometric excess of protein to saturate the DNA? It is not clear what constructs were created to generate the data in Fig. S6l (perhaps a cartoon would help?) In Fig. 5e, please avoid having the red arrows obscure protein-DNA bands.
11. Control of expression by MLDSR binding to motif 1 is discussed. Is MLDSR binding to motif 2 also physiologically relevant?
12. The activation of transcription (Fig. 5m) is not convincing. The reported increase in transcription is marginal. Secondly, the transcription was performed with a phage T7 RNA polymerase, not a bacterial enzyme. It is therefore not clear if this reported increase is even physiologically relevant (i.e., was it due to non-specific interaction between MLDSR and the heterologous polymerase?)
13. The MLDSR-MLDS is said to be encoded in an operon. Under low nitrogen conditions, the increase in MLDS transcription was reported to be ~ 6 -fold (Fig. 3a) while MLDSR was increased >20 -fold (Fig. 7d). Please explain the difference.
14. Line 319. Should be Fig. 8.
15. Fig. 7g-i. Could phenotype of KO be complemented with MLDSR?

16. Fig. 6c-d. The a2-a4 fragment binds DNA, but it is marked in panel c with an x.
17. Line 404. Please include how protein concentration was determined.
18. Line 857. Should be silver, not sliver.
19. Supplemental Fig S6g. Should be Anti-MLDSR, not Anit.

Reviewer #2 (Remarks to the Author):

The manuscript titled "Lipid Droplets Mediate Genomic DNA Localization and Transcription to Promote DNA Stability and Bacterial Survival in Extreme Environments" reports the study on the interactions of the lipid droplet (LD), microorganism lipid droplet small (MLDS) and MLDS regulator (MLDSR) by various techniques including phylogenetic analysis, super-resolution imaging with colocalization analysis, bioinformatics analysis and single molecule methods. The authors have proposed an intriguing mechanistic model on MLDSR regulating either positively or negatively on MLDS for DNA binding and protection, which in turn increased the survival rate of the bacterial cells, RHA1, under extreme conditions such as low nitrogen or UV irradiation. In cytosol, low concentration of MLDSR positively regulates its own transcription and MLDS, while high concentration of MLDSR represses the transcription. In contrast, in LDs, MLDSR was recruited by LDs so that in the cytoplasm, the concentration of MLDSR is maintained in the positive regulation range, illuminating the instrumental role of LDs for the regulation of MLDSR and MLDS. I find the study is fairly comprehensive, the conclusions are mostly solid. However, the quality of this manuscript is compromised by the data presentation. The clarity in the presentation should be improved and additional controls and quantifications are needed in order to be considered acceptance by Nature Communications. My comments are as below.

1. The association between genomic DNA and LDs under various conditions is one important piece of data supporting the model. Therefore I think it's necessary to clarify the imaging and quantification methods from the following few aspects:

1) The authors used two pairs of labeling schemes and microscopies to study the association: (i) staining DNA and LD lipid (with LipidTox) and imaging with SIM, and (ii) staining DNA and LD marker Ro0568 (fused with GFP) and imaging with confocal. It is unclear why colocalization analysis can only be performed in case (ii). Based on SI Fig 1c, Ro0568 stained mostly outside the LipidTox staining region, which seems to be the reason for being used for colocalization analysis, as one can see overlapping DNA and Ro0568 signal, whereas the for case (i) it's not very obvious. Nevertheless, the case (i) is still used for many comparisons, in which arrows are used to indicate association. What is the quantitative criterion to be determined as association? Can you quantify this, and compare with the colocalization analysis?

2) Related to SI Fig 1c, it seems that the colocalization (the overlapping signal) between Ro0568 and LipidTox staining is weaker in MLDS KO case compared to WT. Does that mean the association of the LD marker to LDs is also regulated in some way, which would affect the quantification of DNA/LD colocalization if used as a reference?

3) Need more information about the colocalization analysis in the method. For example, whether this is done in 2D or 3D, and what are the intensity thresholds used in each channel for this analysis? These parameters shall affect the quantification result.

2. Something related to single-molecule pull-down experiment:

1) The images for DNA full-down experiment in Fig 1f are somewhat hard to see. Images for WT-SYTOX and MLDS KO have very different background contrast. I cannot tell the LDs signal in the DIC for WT and the Negative control doesn't have a DIC image.

2) For the second single-molecule assay in figure 2 --- since you also have cy3 labeled DNA, it would be more convincing to repeat the pull-down experiment, and check the colocalization between Cy3 labeled DNA, and Cy5 labeled MLDS.

3. The paragraph between lines 125-131 is very confusing to read. The interpretation does not

strongly reflect SI fig 2d. (1) Authors interpreted the data as “both fusion proteins could bind DNA to the adiposome”, however, in the lane for condition “MLDS-N-H1 + DNA + adiposome” (3&7), majority of DNA is in the solution. (2) It’s also unclear to me why SI fig 2d can lead to the conclusion of “LD-targeting and DNA-binding domains of MLDS were at its N-terminus and C-terminus respectively”, because there is no control to show H1 without fusing to MLDS-N cannot be targeted to adiposome. In fact, this is possible because ADRP seems to be targeted to adiposome without any fusion. I think the most straightforward evidence to support this conclusion is to do the same assay in the MLSD-C (without the N terminal targeting sequence) and demonstrate that MLSD-C binds to DNA but is not in adiposome. (3) What are the three bands in Lane 3?

4. The authors concluded that the mRNA and protein levels of MLDS were significantly reduced in both the deletion and overexpression mutants, whereas in Fig 4f, it appears to me that KO and OE cases have less proteins in general, as evidenced by the bands on the top, especially as position of 95kDa and 55kDa. I think the authors should have a reference band for rigorous comparison.

Minor:

1. Something related to SI Fig. 1

1) Supplementary Fig. 1d, anti-GFP stained multiple bands in addition to the position corresponding to Ro0589. Please explain why this is the case.

2) It is unclear why SI Fig. 1f is used to defined the “purify” of LD, and what’s the definition of the purity for LD as a micro-compartment.

2. Fig 5f. does not have any legend about what x, y axes are, unlike Fig. 5d&3. In addition, in line 219 of the text, it stated $KD(\text{motif } 2) = 5.14 \mu\text{M}$, but Fig. 5f says 5.14 mM.

3. It’s better to also label Motif 2(+), etc as “ssDNA” in SI Fig 6k, if I haven’t misinterpreted.

4. In Fig 6d, a2-a4 truncation can bind DNA, whereas in Fig 6c, a2-a4 is marked as “cannot bind DNA”. I assume this is a typo in Fig 6c? Otherwise the section between lines 253-258 is problematic.

5. The statement that “these findings suggest that oligomerization of the protein was required for its LD location and DNA binding” seems to be too strong. I can see the LD location and DNA binding domains are colocalized with the domains associated with oligomerization. But I don’t see the reasoning leading to the conclusion that MLDSR need to form an oligomer in order to bind DNA and target to LD. For example, the oligomerization of MLDSR could be the end result of binding DNA and targeting LDs, rather than the requirement of these activities.

6. In lines 412-435 it will be more informative to write out the powers of the lasers used.

Based on these, I suggest another review of this manuscript once these corrections and modifications are made.

Reviewer #3 (Remarks to the Author):

This manuscript focuses on the function of two lipid droplet proteins in *Rhodococcus* bacteria. Droplets are known to exist in these cells and proteomics of the droplets has previously been performed. This paper is about two droplet proteins, MLDS (microorganism lipid droplet small) and its transcriptional regulator MLDSR. The authors first show by fluorescence microscopy and fractionation that DNA tends to normally aggregate close to droplets, and they show that MLDS is required for this behavior. They next show that MLDS protects DNA from damage under stress conditions. The second half of the paper is about MLDSR. The authors show nicely that MLDSR

regulates the transcription of MLDS. They identify two DNA binding sites, one in the operator, the other in the promoter, and they narrow down the region in MLDSR necessary and sufficient for DNA binding and droplet binding. They show that droplets serve to titrate the free concentration of MLDSR that regulates transcription of both MLDS and MLDSR.

Overall, this is a detailed dissection of the function of two droplet proteins and the role that droplets play in their regulation. Very little is known about the function of droplets in bacteria, and this adds a lot to this field. The work suggests that a major function of both prokaryotic and eukaryotic droplets is the regulation of some basic nucleic acid activities.

That said, there are some issues that the authors should address:

(1) Figure 1f: droplets are nearly impossible to visualize!! Why is the SYTOX background so much higher in the WT vs the KO strain?

(2) Fig 2d: don't the authors think that specific protein-DNA interactions occur at a molar ratio of 50? Everything else may be nonspecific. At a ratio of 220, my calculations (based on estimate of diameter of the protein and length of DNA per base) show total saturation of DNA with MLDS.

(3) Fig. 2i and others: I find the nomenclature MLDS-N (etc) confusing. To me it denotes the protein minus the amino terminus. Consider using superscript instead throughout?

(4) Fig 2h: some cartoons like this one are confusing - it looks as if both samples are mixed before centrifuging. Is this what is meant?

(5) Fig. 2i and others: Are we comparing equal % of adiposomes and solution? This is not clear from methods or legend.

(6) Fig 3f: why doesn't the KO+MLDS FULLY rescue? You are putting back exactly what was missing. This needs an explanation.

(7) Fig. 3j: is it fair to compare the tail moments if the signal (fluorescence brightness) is so different? Could not some of the measurement be affected by detection sensitivity?

(8) I'd like to see how much MLDS is in the droplet vs. cytosol fractions.

(9) Related to above, Fig. 4b: If equal protein are loaded, one gets a false idea of distribution, as droplets have so little protein. The cytosol may have 50% of ro02105, for example, while the specific activity is very much higher in the droplet fraction.

(10) Fig 5a: This is most confusing! There are 2 ~40-bp fragments shown in the bottom of the panel. But only one is run out in 5b, right. I'm confused about these two small fragments.

(11) Fig 5d and 5e: why wasn't Motif 2 by itself analyzed?

(12) Text regarding Fig 3c: is it fair to score proximity in extremely low nitrogen as the cells are so much shorter? The DNA has nowhere to go other than by a droplet.

(13) lines 221-222: the palindrome is not obvious from this sequence; please make this clearer

(14) line 257: that alpha2 shares the two functions (droplet localization and DNA binding) is an overinterpretation. That sequence may be required for a general folding function of the protein rather than the more specific functions discussed.

(15) line 353: confusing: "composition of the membrane layer surrounding them" as in some

cases there is an absence of a membrane.

(16) General comment: The DNA is not tightly adhering to droplets in the micrographs, but is spread out such that much of the DNA is 1-2 micrometers away. Why is there such a large effect on DNA stability, especially after UV damage? Do the authors believe that droplets can stabilize DNA even when the DNA is not that close to the droplets?

Reviewer #4 (Remarks to the Author):

The authors of the manuscript entitled "Lipid Droplets Mediate Genomic DNA Localization and Transcription to Promote DNA Stability and Bacterial Survival in Extreme Environments" studied the protective role of lipid droplets to genomic DNA. The authors concluded that MLDS protein and MLDSR are involved in the process by binding DNA and providing a controlling method for the lipid droplets.

Although the study has merit and seems to have been competently carried out, the authors never discuss aspects that may have greatly influenced the binding of DNA to the lipid droplets such as the lipidic nature of the LD, in particular the phospholipids in the monolayer membrane that will contact with the DNA, and the inherent polarity of the DNA chain. Both electrostatic and hydrophobic/hydrophilic interactions should have been considered in the interpretation of the results. The authors may decide that they only wanted to study the role of proteins in LD but the lipid character of LD must play a decisive role in the binding of DNA and other molecules. The authors should at least discuss this aspect in the discussion and also the chemical properties of both lipids and proteins in LD, and DNA.

The association of genomic DNA with LDs is not clear by the SIM and confocal microscopy images since what the white arrows are pointing to varies in the several images. Besides, the name of the "green" dye should be Hoechst® and not Hochest. However, according to the manufacturer of this dye, the DNA should be stained blue. Green emission has been observed when the dye suffers photobleaching as described by Żurek-Biesiada et al. *J Fluoresc.* 2014; 24(6): 1791–1801. What is the explanation for the green colour e.g. in SFig 1?

The work is also very dependent on microscopy images. For example, L492 "Fluorescence intensity from SYTOX staining and overexpressed GFP-tagged LD markers was quantified with Image J software." However, intensity measurements in images are not a "bullet proof" technique since the amount of light reaching the sample on each image may vary. The authors should have measured the intensity peaks e.g. by fluorometry or flow cytometry or present the results as intensity ratios.

Other comments (L=Line):

- L8: "through the major LD protein, microorganism lipid droplet small (MLDS), which (...)"
The phrase should be "through the major LD protein, microorganism lipid droplet small (MLDS) protein, which". In fact, "microorganism lipid droplet small" does not make much sense in many of the phrases in the text. This also appears in L35.

L30: "Rhodococcus opacus PD630 (PD630) and Rhodococcus jostii RHA1 (RHA1) are known to contain more LDs and TAG content than any other bacteria identified to date."
This was true in 2001 but is it still?

L84: "We then utilize" should be "We then utilized"

L115: "DNA and proteins were then incubated with the adiposomes, and then the reaction was centrifuged to separate the adiposomes from the reaction solution (Fig. 2h)."
To which reaction are the authors referring to? DNA and protein incorporation in LD should not be

the result of a reaction.

L509: "The cells were treated with UV exposure" should be "The cells were exposed to UV light".

Besides, the authors refer that the cells grew to $OD \sim 2$ in liquid medium and were then cultured on LB solid medium. No information on the age of cells (stationary or exponential phase), initial number of cells used to inoculate the solid medium, and time during which the cells were cultured before being exposed to UV light is referred. It is known that number of cells and age of culture influence greatly the survival to UV light.

L544: How were the cells homogenized? The method may influence the amount and quality of LD recovered.

L547: "To prepare the LD protein sample, 1 mL of chloroform:acetone (1:1, v/v) was added. LD proteins were mixed with 2×SDS sample buffer and denatured at 95°C for 5 min."

Were the proteins from the LD i) extracted and used whilst others were denatured or ii) extracted and denatured?

Reviewer #1 (Remarks to the Author):

This manuscript reports that the lipid droplet in *Rhodococcus jostii* binds and protects genomic DNA. This binding requires the MLDS protein. It is also reported that expression of MLDS depends on the regulator MLDSR.

The role of the lipid droplet in organizing and protecting genomic DNA is interesting, and the contribution of MLDS to DNA binding is convincing.

The proposed mechanism by which MLDSR controls MLDS is not quite as clear (particularly the activation), as detailed below (items 10-12):

1. Perhaps it is the reproduction, but I cannot see any labeling in Fig. 1f.

Response: We thank the reviewer's critical comment. We are sorry that we cannot explain the data very well because our collaborator of this experiment, professor Yongfang Zhao, passed away not long ago and her lab was closed. In addition, the data did not add essential information. Therefore, we decided to remove Fig. 1f from the revised manuscript.

2. Please define abbreviations on first use (e.g., ADRP, line 126; DOPC, line 448).

Response: We thank the reviewer's useful comment. We have defined these abbreviations on first use in the revised manuscript.

3. Line 107. In Fig. 2f, does MLDS-C' correspond to just the lysine-rich region with the lysines replaced? Please clarify. Was full-length MLDS in which lysines were mutated tested? Or the N-terminal domain of MLDS? As it stands, this assay indicates that the lysine-rich C-terminus is involved in DNA binding, but it does not rule out that the N-terminal domain contributes.

Response: We thank the reviewer's critical comments. Yes, MLDS-C' is just the lysine-rich region with the lysines replaced. We did not test the full-length MLDS with lysine mutations. But we performed adiposome-binding assay with MLDS-N and found that compared with full-length MLDS, MLDS N-terminus could not bind DNA to adiposomes (Fig. 2i and Fig. S2c). We also overexpressed MLDS-N in MLDS KO cells and found that it could not rescue phenotype of MLDS KO (Fig. S1b), suggesting that MLDS-N is lack of DNA binding ability. We included a new experiment as Fig. S2b in which MLDS N-terminus was not able to bind DNA in EMSA. In addition, our previous paper has revealed that MLDS locates on LDs by its

N-terminus (Ding, Y. et al. Identification of the major functional proteins of prokaryotic lipid droplets. *J Lipid Res* 53, 399-411 (2012)).

4. Line 125/Fig. 2i/Fig. S2d. Some DNA is associated with adiposomes in the MLDS-N mutant as well as in ADRP. Would that suggest that DNA binding may occur either non-specifically or through the MLDS N-terminal domain?

Response: We thank the reviewer's critical comments. Through all our experiments, we think that those associations are non-specific binding by incomplete washing of adiposomes after the reactions. In fact, DNA signal was not detected on adiposomes in some other controls (Fig. 2j, lanes 2 and 7, and Fig. S3f, lane 3). Importantly, we can judge whether a protein binds DNA to adiposomes or not through 1) comparing the amount of DNA in adiposome fraction and solution fraction, and 2) checking whether DNA is decreased in solution fraction compared with original DNA (Fig. 2i and S2d, lane 1).

5. Line 139. It is said that no DNA signal is seen with adiposomes and JLP, however, a weak signal is seen in Fig. S3f, lane 1? More careful wording may be required.

Response: We thank and agree the reviewer's useful suggestion. We changed "no DNA signal" to "no or little DNA signal" in the revised manuscript.

6. Line 168/Fig. 3f. Why is growth phenotype only partially rescued by expression of MLDS?

Response: We thank the reviewer's critical question. In fact, we do not know the answer. Our explanation is that since MLDS deletion also enlarges LD size (Ding, Y. et al. *J Lipid Res* 53, 399-411 (2012)), it may be a multifunctional protein. Therefore, deletion of MLDS may change bacteria dramatically and re-expression may not be able to 100% recover the growth phenotype.

7. Fig. 3g. Can the authors comment on why the comet assay only shows a difference after 24 h? No difference is seen after 48 h or longer.

Response: We thank the reviewer's question. In fact, we have not dissected the detailed mechanism of MLDS-mediated DNA protection. We speculate that LDs stabilize DNA during DNA replication since cells are growing at 24 h time point while cells turn into stationary phase at 48 h (Fig. 3f).

8. Fig. 3h-i. Why was MLDS KO strain complemented with full-length MLDS not included?

Response: We thank the reviewer's critical comment. We included a new experiment in Fig. 3i in the revised manuscript.

9. Line 198. What was the justification for selecting this particular band? Please mention in the main text how this protein was identified. Is the operon predicted or confirmed? "orchestrated" may not be the best term here.

Response: We thank the reviewer's useful comment. In the main text we replaced "In a comparison of total LD proteins between WT and MLDS KO cells, a band" by "When LD protein profiles of WT and MLDS KO were compared using SDS-PAGE, the band containing MLDS was greatly reduced while the other band was appeared in the MLDS KO mutant (Fig. 4a, red arrow). The mass spectrometry (MS) analysis showed that the band contained a putative transcriptional regulator, *RHA1_ro02105*. The N-terminus of this protein was predicted by the START database to contain a xenobiotic response element (XRE) helix-turn-helix (HTH) DNA-binding motif (Supplementary Fig. 5a). The *RHA1_ro02105* gene region overlaps slightly with the *mlds* gene in the genome and it is predicted that the two genes were in the same operon (Supplementary Fig. 5a)." In addition, the data in this study prove that the cis-element of both the two genes is same and in their upstream DNA sequence. We have re-described the result in the revised manuscript.

10. The binding assays for MLDSR (Fig. 5) are confusing. According to Methods, a single site binding model was used for determination of K_d from SPR assays, however, the DNA in Fig. 5d contains two sites, so this is not the correct binding model. The text (line 219) indicates a 5.14 microM K_d for motif 2, but Fig. 5f says millim. According to Methods, EMSA was performed with 400-500 ng DNA in a 15 microL reaction, corresponding to ~0.5-1 microM DNA (depending on length). If the K_d for 43 bp DNA is 62 nM (Fig. 5d), that means EMSAs were performed under stoichiometric conditions; if this is so, then why does it require a stoichiometric excess of protein to saturate the DNA? It is not clear what constructs were created to generate the data in Fig. S6l (perhaps a cartoon would help?) In Fig. 5e, please avoid having the red arrows obscure protein-DNA bands.

Response: We thank the reviewer's critical comments.

It's true that we used a single site binding model to determine the K_D of motif (1+2) in Fig. 5d. Because even though the motif (1+2) contains two sites, these two sites are adjacent, which indicates that motifs 1 and 2 may be cooperative for MLDSR binding. Furthermore, the results showed that MLDSR could form oligomers to bind DNA (Fig. S7). Based on the putative binding cooperatively of transcriptional regulator

oligomers (textbook GENES), the protein binding to one site may increase the affinity of binding to the adjacent site. Thus the motif (1 + 2) should not be regarded as two independent sites. Thus, we think that the result represents apparent affinity for motif (1+2).

The reviewer is correct that it is a mistake to mark the Kd of motif 2 as 5.14 milliM in Fig.5f. We corrected it in the revised manuscript. We very appreciate the reviewer to point it out.

We calculated the Kd of 43 bp DNA in EMSA. Based on Fig. 5b, it was saturated when the molar ratio of MLDSR/DNA is about 10 and the amount of MLDSR-DNA complex is 12-fold than that of free DNA (grey level analysis by ImageJ). Thus, $[MLDSR-DNA] = 12[DNA]_e$, $[DNA]_i = 13[DNA]_e$, and $[MLDSR]_i = 10 \times 13[DNA]_e = 130[DNA]_e$. ($[DNA]_e$ and $[DNA]_i$ represents the concentration of DNA at equilibrium and initiation, respectively. $[MLDSR]_e$ and $[MLDSR]_i$ represent the concentration of DNA at equilibrium and initiation, respectively.) Based on the symmetrical property of motif 1 and motif 2 (Fig. 5g) and MLDSR oligomerization property (Fig. S7), we speculate that a molecular of motif (1 + 2) can bind 4~8 MLDSR (mean is 6). Therefore, $[MLDSR]_e = [MLDSR]_i - 6[MLDSR-DNA] = 130[DNA]_e - 6 \times 12[DNA]_e = 58[DNA]_e$. In the assay, we know that $[DNA]_i$ is 1.2 microM. So $[DNA]_e$ is about 92 nM. Altogether, $Kd = ([MLDSR]_e * [DNA]_e) / [MLDSR-DNA] = (58[DNA]_e * [DNA]_e) / 12[DNA]_e = 4.8[DNA]_e = 442$ nM. With our rough calculation, it seems that the Kd 62 nM in SPR assay and the Kd 442 nM in EMSA fall into a same range of affinity. Since detection by EB staining is not a good quantification method, and SPR assay is more sensitive and accurate, we used the SPR assay for quantitative analysis and the EMSA for qualitative analysis to identify the interaction between DNA and MLDSR.

These mutated DNA motif probes are from the synthesized and annealed oligonucleotides (< 60 bp). We included this information in Methods in the revised manuscript.

We removed the red arrows from Fig. 5b and c in the revised manuscript.

11. Control of expression by MLDSR binding to motif 1 is discussed. Is MLDSR binding to motif 2 also physiologically relevant?

Response: We thank the reviewer's critical question. We think that binding of MLDSR to motif 2 is physiologically relevant. First, the binding affinity of MLDSR to motif 2 was 5.14 μ M, which seems to be a physiological binding. Second, according to the results (Fig. S6m and n), when T7 promoter and motif 1 were not adjacent, the repression of MLDSR in transcription was reduced non-detectable. It not only suggests that the repression is occurred at transcription initiation but also indicates that MLDSR may regulate transcription by affecting RNA polymerase binding to the promoter. In addition, our data suggest that motif 2 is a promoter region (Fig. 5h-j). Therefore, we proposed that MLDSR regulates transcription by binding motif 2 in bacteria physiologically.

12. The activation of transcription (Fig. 5m) is not convincing. The reported increase in transcription is marginal. Secondly, the transcription was performed with a phage T7 RNA polymerase, not a bacterial enzyme. It is therefore not clear if this reported increase is even physiologically relevant (i.e., was it due to non-specific interaction between MLDSR and the heterologous polymerase?)

Response: We thank the reviewer's critical comments.

First, we claimed that MLDSR could regulate transcription positively because when MLDSR was knockout MLDS expression was decreased (Fig. 4e, f). If MLDSR just represses MLDS expression, MLDS expression could not be reduced when MLDSR knockout.

Second, it is true that the reported increase in transcription is not dramatic, but it is significant and reproducible (Fig. 5m and 6h, last 3 lanes). The result showed that MLDS could still be expressed when MLDSR was knockout (Fig. 4e, f). Thus, we think that the marginal increase is because MLDSR is not the only one protein for the transcriptional regulation. RNA polymerase may work by itself or with other proteins. Third, we do not know detailed mechanism yet, but based on the results (Fig. 5l, m and 6i) we speculate when MLDSR concentration is high, MLDSR represses transcription by binding both motifs to affect RNA polymerase binding to the promoter. When MLDSR concentration is low, according to the binding affinity of MLDSR to motif 1 and motif 2 (motif 1 > motif 2) (Fig. 5e, f), MLDSR may only bind to motif 1 and promote RNA polymerase binding to the promoter.

In addition, previous works revealed that several transcriptional regulators could function as both repression and activation, for example, lambda repressor (textbook GENES), which indicates the possibility of MLDSR functions as both repression and activation.

All together, we agree that the mechanism of transcription activation is required further dissection, but we do think that the transcription activation is physiologically relevant.

13. The MLDSR-MLDS is said to be encoded in an operon. Under low nitrogen conditions, the increase in MLDS transcription was reported to be ~6-fold (Fig. 3a) while MLDSR was increased >20-fold (Fig. 7d). Please explain the difference.

Response: We thank the reviewer's critical comments. The experiments were performed triplicates and the results were similar. We speculate that it is due to the mRNA stability. Because bacterial mRNA is degraded in a 3'-5' way usually (Wikipedia of "Degradosome" and Górná, Maria W.; Carpousis, Agamemnon J.; Luisi, Ben F. (2012-05-01). "From conformational chaos to robust regulation: the structure and function of the multi-enzyme RNA degradosome". *Quarterly Reviews of Biophysics*. 45 (2): 105–145.) Since the mRNA of MLDS is at the 3' terminus of the

MLDSR-MLDS mRNA, the mRNA of MLDS is easier to be degraded. In addition, according the Western blot results (Fig. 3b and 7e), it is found that the protein level of MLDS and MLDSR are increased under low nitrogen conditions. However, the increase is about 2-3 fold, which suggests the mRNA indeed is degraded. Thus, we think that the difference dues to the mRNA degradation.

14. Line 319. Should be Fig. 8.

Response: We thank the reviewer for pointing out it and corrected the mistake in the revised manuscript.

15. Fig. 7g-i. Could phenotype of KO be complemented with MLDSR?

Response: We thank the reviewer's critical comment. We do not know whether MLDSR can rescue the MLDSR KO phenotype. Because the phenotypes of MLDSR KO and MLDSR OE were similar, and the expression of MLDSR is hard to be controlled. Furthermore, the transcriptional regulation activity of MLDSR was also mediated by LDs (Fig. 6) and the amount of LDs and the level of LDs involving in the process are also hard to be controlled. Thus, it can be very difficult to rescue the MLDSR KO by performing the complementary experiments. But according to the results, we know that MLDSR knockout and overexpression can both reduce MLDS expression (Fig. 4e, f) and their phenotypes are similar to the phenotype of MLDS KO (Fig. 3 and 7), which suggests that MLDSR plays role in bacterial physiological process. Thus, we think that the conclusion that MLDSR regulates MLDS expression to control genomic DNA localization and stability is convincing.

16. Fig. 6c-d. The a2-a4 fragment binds DNA, but it is marked in panel c with an x.

Response: We thank the reviewer for pointing out it and corrected the mistake in the revised manuscript.

17. Line 404. Please include how protein concentration was determined.

Response: We thank the reviewer's useful suggestion and included it in the revised manuscript.

18. Line 857. Should be silver, not sliver.

Response: We thank the reviewer for pointing out it and corrected the mistake in the revised manuscript.

19. Supplemental Fig S6g. Should be Anti-MLDSR, not Anit.

Response: We thank the reviewer for pointing out it and corrected the mistake in the revised manuscript.

Reviewer #2 (Remarks to the Author):

The manuscript titled “Lipid Droplets Mediate Genomic DNA Localization and Transcription to Promote DNA Stability and Bacterial Survival in Extreme Environments” reports the study on the interactions of the lipid droplet (LD), microorganism lipid droplet small (MLDS) and MLDS regulator (MLDSR) by various techniques including phylogenetic analysis, super-resolution imaging with colocalization analysis, bioinformatics analysis and single molecule methods. The authors have proposed an intriguing mechanistic model on MLDSR regulating either positively or negatively on MLDS for DNA binding and protection, which in turn increased the survival rate of the bacterial cells, RHA1, under extreme conditions such as low nitrogen or UV irradiation. In cytosol, low concentration of MLDSR positively regulates its own transcription and MLDS, while high concentration of MLDSR represses the transcription. In contrast, in LDs, MLDSR was recruited by LDs so that in the cytoplasm, the concentration of MLDSR is maintained in the positive regulation range, illuminating the instrumental role of LDs for the regulation of MLDSR and MLDS. I find the study is fairly comprehensive, the conclusions are mostly solid. However, the quality of this manuscript is compromised by the data presentation. The clarity in the presentation should be improved and additional controls and quantifications are needed in order to be considered acceptance by Nature Communications. My comments are as below.

1. The association between genomic DNA and LDs under various conditions is one important piece of data supporting the model. Therefore I think it's necessary to clarify the imaging and quantification methods from the following few aspects:

1) The authors used two pairs of labeling schemes and microscopies to study the association: (i) staining DNA and LD lipid (with LipidTox) and imaging with SIM, and (ii) staining DNA and LD marker Ro0568 (fused with GFP) and imaging with confocal. It is unclear why colocalization analysis can only be performed in case (ii). Based on SI Fig 1c, Ro0568 stained mostly outside the LipidTox staining region, which seems to be the reason for being used for colocalization analysis, as one can see overlapping DNA and Ro0568 signal, whereas the for case (i) it's not very obvious.

Nevertheless, the case (i) is still used for many comparisons, in which arrows are used to indicate association. What is the quantitative criterion to be determined as association? Can you quantify this, and compare with the colocalization analysis?

Response: We thank the reviewer's critical comments. LDs are spherical structure with neutral lipid core covered by phospholipid membrane and proteins. Based on our study, DNA did not directly contact with LD phospholipids and neutral lipids (Fig. 2i and 2j, lane 2). In between DNA and LD neutral lipids there were phospholipid membrane and proteins. In our finding, protein MLDS was the link to connect LDs and DNA. Thus, no overlapping signal between DNA and LD neutral lipids (LipidTox) should be detected. That is why it is hard to quantify the overlapping between DNA and LDs (neutral lipid staining) in SIM images. Using double staining of DNA and neutral lipids, we could only compare the proximity of DNA and LD neutral lipids between MLDS WT and MLDS KO cells. To verify the finding of double staining experiment, we performed the colocalization assay. In the study, we used the SIM observation in three places mainly (Fig. 1c, 3c and 7a). When we wanted to compare the association of DNA and LDs under various conditions quantitatively, we always performed the colocalization assays (Fig. 1d-e, 3d-e, and 7b-c). Since LDs in bacteria are very small and average size is about 0.3 μm diameter, visualizing bacterial LDs is difficult. Two experimental designs can be complementary advantage as well as verify each other. Furthermore, on top of imaging the results from biochemical and molecular experiments including *in vitro* assays sufficiently support our conclusion.

2) Related to SI Fig 1c, it seems that the colocalization (the overlapping signal) between Ro0568 and LipidTox staining is weaker in MLDS KO case compared to WT. Does that mean the association of the LD marker to LDs is also regulated in some way, which would affect the quantification of DNA/LD colocalization if used as a reference?

Response: We thank the reviewer's critical comments. We are sorry for the confusion. LDs are spherical structure with neutral lipid core and phospholipid membrane. In general, LD proteins do not directly contact with LD neutral lipids. Therefore, the overlapping signal between proteins and LD neutral lipids should not be obvious. In current study, we did not try to use overlapping signal to identify the association of proteins to LDs. In 2D image, LDs were red round dots with LipidTox staining while the Ro05869-GFP was "ring" structure, which is in agreement with LD physiology. In addition, according to the results (SI Fig. 1d), Ro05869 expression and the protein on LDs were similar in WT and MLDS KO cells, and almost all the signals detected by anti-GFP were in LD fractions. The experiment in SI Fig. 1c was designed to identify that Ro05689 is a LD-associated protein and the weaker signals in MLDS KO cells (the Ro05689-GFP signal and lipidTox signal were both weaker) may be due to the weaker intensity of lasers. In colocalization assay, the almost same intensity of lasers

was used and the green signal of Ro05869-GFP was similar in WT and MLDS KO cells (Fig. 1d).

Above all, we think that Ro05869 can be a specific LD marker for colocalization assay.

3) Need more information about the colocalization analysis in the method. For example, whether this is done in 2D or 3D, and what are the intensity thresholds used in each channel for this analysis? These parameters shall affect the quantification result.

Response: We thank the reviewer's useful suggestion. We have added the information in the method in the revised manuscript. They are 2D images. In WT cells cultured in MSM with 0.5 g/L NH₄Cl for control, it was used the mean threshold of 1,440 for SYTOX signal and the mean threshold of 670 for RHA1_ro05869-GFP signal. In MLDS KO cells, it was used the mean threshold of 1,330 for SYTOX signal and the mean threshold of 610 for RHA1_ro05869-GFP signal. In MLDSR KO cells, it was used the mean threshold of 1,150 for SYTOX signal and the mean threshold of 800 for RHA1_ro05869-GFP signal. In WT cells cultured in MSM with 0.1 g/L NH₄Cl, it was used the mean threshold of 1,400 for SYTOX signal and the mean threshold of 760 for RHA1_ro05869-GFP signal. In MLDSR OE cells, it was used the mean threshold of 1,730 for SYTOX signal and the mean threshold of 1,150 for MLDSR-GFP signal. The colocalization analysis was modified from the method described in the paper (Bei-Bei Chu et. al, Cholesterol Transport through Lysosome-Peroxisome Membrane Contacts. Cell, Volume 161, Issue 2, 2015, Pages 291-306).

2. Something related to single-molecule pull-down experiment:

1) The images for DNA full-down experiment in Fig 1f are somewhat hard to see. Images for WT-SYTOX and MLDS KO have very different background contrast. I cannot tell the LDs signal in the DIC for WT and the Negative control doesn't have a DIC image.

Response: We thank the reviewer's critical comment. We are sorry that we cannot explain the data very well because our collaborator of this experiment, professor Yongfang Zhao, passed away not long ago and her lab was closed. In addition, the data did not add essential information to the study. Therefore, we decided to remove Fig. 1f from the revised manuscript.

2) For the second single-molecule assay in figure 2 --- since you also have cy3 labeled DNA, it would be more convincing to repeat the pull-down experiment, and check the colocalization between Cy3 labeled DNA, and Cy5 labeled MLDS.

Response: We thank the reviewer's critical comments, but we are sorry for that we cannot repeat the assay, because our collaborator of the assay, professor Yongfang Zhao, passed away not long ago and her lab was closed. However, through comparing the signal with and without DNA in Figure 2b and 2c, it is clear and significant to show the interaction between MLDS and DNA.

3. The paragraph between lines 125-131 is very confusing to read. The interpretation does not strongly reflect SI fig 2d. (1) Authors interpreted the data as "both fusion proteins could bind DNA to the adiposome", however, in the lane for condition "MLDS-N-H1 + DNA + adiposome" (3&7), majority of DNA is in the solution. (2) It's also unclear to me why SI fig 2d can lead to the conclusion of "LD-targeting and DNA-binding domains of MLDS were at its N-terminus and C-terminus respectively", because there is no control to show H1 without fusing to MLDS-N cannot be targeted to adiposome. In fact, this is possible because ADRP seems to be targeted to adiposome without any fusion. I think the most straightforward evidence to support this conclusion is to do the same assay in the MLSD-C (without the N terminal targeting sequence) and demonstrate that MLSD-C binds to DNA but is not in adiposome. (3) What are the three bands in Lane 3?

Response: We thank the reviewer's critical comments.

(1) It is true that majority of DNA is in the solution. But we can judge whether a protein binds DNA to adiposomes or not through 1) comparing the amount of DNA in adiposome fraction and solution fraction, 2) comparing the amount of DNA in adiposome fraction between target protein and control protein, and 3) checking whether DNA is decreased in solution fraction compared with original DNA (lane 1). According the result, the DNA of lane 3 was less than that of lane 7, but more than that of lane 2, and the DNA of lane 7 was less than that of lane 1. Thus, MLDS-N-H1 was able to bind DNA to adiposomes.

(2) We are sorry for the confusion. It is reported that histones cannot locate on LDs by themselves, an anchor protein Jabba is necessary for the location (Li, Z. et al. Lipid droplets control the maternal histone supply of Drosophila embryos. *Curr Biol* 22, 2104-13 (2012)). We mentioned the information in introduction in the manuscript, "Furthermore, in another study, histones were found localized to LDs via the anchor protein Jabba in Drosophila". ADRP is a LD resident protein in mammalian cells (we included the information in the revised manuscript). Thus, ADRP can target to adiposomes (Wang, Y. et al. Construction of Nano-Droplet/Adiposome and Artificial Lipid Droplets. *ACS Nano* (2016)) and histone H1 could not. Therefore, the targeting of MLDS-N-H1 to adiposomes suggests that MLDS-N is the targeting domain. In fact, our previous work shows that MLDS-N contains the LD-targeting domain (Ding, Y. et al. Identification of the major functional proteins of prokaryotic lipid droplets. *J Lipid Res* 53, 399-411 (2012)). In current study, MLDS-N was found to be localized to adiposomes (Fig. 2i, lane 3 and 4). The result in SI Fig. 2d further supports the

conclusion. The conclusion that MLDS-C is the DNA binding domain has been represented in the study (Fig. 2). The result in SI Fig. 2d further supports the conclusion.

(3) We guess that because the fusion protein is instable. Full-length MLDS-N-H1 is degraded into several pieces. Based on the prediction of molecular weight, the highest band of the three bands is putative full-length MLDS-N-H1.

4. The authors concluded that the mRNA and protein levels of MLDS were significantly reduced in both the deletion and overexpression mutants, whereas in Fig 4f, it appears to me that KO and OE cases have less proteins in general, as evidenced by the bands on the top, especially as position of 95kDa and 55kDa. I think the authors should have a reference band for rigorous comparison.

Response: We thank the reviewer's critical comments. In fact, it is hard to find a robust reference band in bacterial LD fraction. Though Ro05869 is considered, we think that comparing the total proteins of LD fraction in general is better. In figure 4f, it is true that the proteins of KO lane were less than that of WT lane. But the decrease of MLDS was more significant and we repeated the experiment in SI F5d. The proteins of KO lane and WT lane were similar, and MLDS was also decreased dramatically. In figure 4f, the proteins of OE lane were similar with that of WT lane because of the large amount of MLDSR-GFP. Thus, protein levels of MLDS were significantly reduced in both the deletion and overexpression mutants.

Minor:

1. Something related to SI Fig. 1

1) Supplementary Fig. 1d, anti-GFP stained multiple bands in addition to the position corresponding to Ro0589. Please explain why this is the case.

Response: We thank the reviewer's question. We guess that the multiple bands may be non-specific bands or degraded bands. Because Ro05869 was overexpressed and almost all proteins locate on LDs, it is possible that the protein was degraded. However, according to the results (SI Fig. 1c, d), almost all Ro05869 were around/on LDs (green signal or bands), which suggests that the protein still is a specific marker protein of LDs for co-localization analysis.

2) It is unclear why SI Fig. 1f is used to defined the "purify" of LD, and what's the definition of the purity for LD as a micro-compartment.

Response: We thank the reviewer's question and we are sorry for the confusion. In fact, we would like to determine the purity of LD fraction in cell fractionation analysis. The ways we determine the purity of LD fraction include 1) analysis of

organelle marker proteins to determine enrichment of LDs and contamination of other cell fractions, 2) comparison of protein profiles between LDs and other cell fractions. Based on this principle, we determined the purity of LD fraction by three methods: 1) Because GFP is not a LD-associated protein and localized in cytosol, GFP was used as a negative control. The results showed that indeed GFP was only in cytosol fraction, not LD fraction. 2) It is reported previously that Ro05869 is a LD-associated protein, thus we use it as a LD marker. 3) The protein profile of LDs is different from the other cell fractions. Above all, the result in SI Fig. 1f suggested that the LD fraction was relative purer and contained a little of contamination with other cell fractions.

2. Fig 5f. does not have any legend about what x, y axes are, unlike Fig. 5d&3. In addition, in line 219 of the text, it stated $KD(\text{motif } 2) = 5.14 \mu\text{M}$, but Fig. 5f says 5.14 mM.

Response: We thank the reviewer's correction. They are mistakes indeed and we corrected them in the revised manuscript.

3. It's better to also label Motif 2(+), etc as "ssDNA" in SI Fig 6k, if I haven't misinterpreted.

Response: We thank the reviewer's useful suggestion. We included it in the revised manuscript.

4. In Fig 6d, a2-a4 truncation can bind DNA, whereas in Fig 6c, a2-a4 is marked as "cannot bind DNA". I assume this is a typo in Fig 6c? Otherwise the section between lines 253-258 is problematic.

Response: We thank the reviewer's useful comment. It is our mistakes in Fig. 6c and we corrected it in the revised manuscript.

5. The statement that "these findings suggest that oligomerization of the protein was required for its LD location and DNA binding" seems to be too strong. I can see the LD location and DNA binding domains are colocalized with the domains associated with oligomerization. But I don't see the reasoning leading to the conclusion that MLDSR need to form an oligomer in order to bind DNA and target to LD. For example, the oligomerization of MLDSR could be the end result of binding DNA and targeting LDs, rather than the requirement of these activities.

Response: We thank and agree with the reviewer's useful comments. We changed it in the revised manuscript as "these findings suggest that MLDSR could locate on LDs and bind DNA as oligomers".

6. In lines 412-435 it will be more informative to write out the powers of the lasers used.

Response: We thank the reviewer's useful suggestion and added them in the revised manuscript. "The powers of the lasers are 500 mW (561 nm), 500 mW (488 nm), and 600 mW (405 nm), respectively. The %T numbers are 1.0~10.0 for 561 nm and 488 nm lasers and 31.3 for 405 nm laser."

Reviewer #3 (Remarks to the Author):

This manuscript focuses on the function of two lipid droplet proteins in *Rhodococcus* bacteria. Droplets are known to exist in these cells and proteomics of the droplets has previously been performed. This paper is about two droplet proteins, MLDS (microorganism lipid droplet small) and its transcriptional regulator MLDSR. The authors first show by fluorescence microscopy and fractionation that DNA tends to normally aggregate close to droplets, and they show that MLDS is required for this behavior. They next show that MLDS protects DNA from damage under stress conditions. The second half of the paper is about MLDSR. The authors show nicely that MLDSR regulates the transcription of MLDS. They identify two DNA binding sites, one in the operator, the other in the promoter, and they narrow down the region in MLDSR necessary and sufficient for DNA binding and droplet binding. They show that droplets serve to titrate the free concentration of MLDSR that regulates transcription of both MLDS and MLDSR.

Overall, this is a detailed dissection of the function of two droplet proteins and the role that droplets play in their regulation. Very little is known about the function of droplets in bacteria, and this adds a lot to this field. The work suggests that a major function of both prokaryotic and eukaryotic droplets is the regulation of some basic nucleic acid activities.

That said, there are some issues that the authors should address:

(1) Figure 1f: droplets are nearly impossible to visualize!! Why is the SYTOX background so much higher in the WT vs the KO strain?

Response: We thank the reviewer's critical comment. We are sorry that we cannot explain the data very well because our collaborator of this experiment, professor Yongfang Zhao, passed away not long ago and her lab was closed. In addition, the

data did not add essential information. Therefore, we decided to remove Fig. 1f from the revised manuscript.

(2) Fig 2d: don't the authors think that specific protein-DNA interactions occur at a molar ratio of 50? Everything else may be nonspecific. At a ratio of 220, my calculations (based on estimate of diameter of the protein and length of DNA per base) show total saturation of DNA with MLDS.

Response: We thank the reviewer's critical question. We think that all bindings are specific. Fig. 2d shows that the binding was gradually increased since free monomer DNA was decreased with increased protein. We have to confess that the diffused gel shifting indeed gave a confused image that seems to be a step shift at molar ratio of 50. The reviewer is right that the binding between DNA and protein MLDS reached saturation at a ratio of 220. Our calculation as following: according to the experiment (Fig. 2d), the 2.5 kb DNA was used for the EMSA and the molecular weight of protein-DNA complex corresponds to the 22.5 kb DNA (the marker is 23.5 kb in Fig. 2d), which suggests that the molecular weight of protein in protein-DNA complex corresponds to the 20 kb DNA. The molecular weight of GST-MLDS is about 60 kDa and the molecular weight of 1 bp DNA is about 660 Da. It means that 1 kb DNA corresponds to 11 GST-MLDS and 20 kb DNA corresponds to 220 GST-MLDS. Thus, the calculations suggest that the 2.5 kb DNA can bind to 220 GST-MLDS, which is consistent with the result. The data indicates that one GST-MLDS is bound to each 12 bp DNA because MLDS binds to DNA without DNA sequence specificity. The estimated diameter of protein is about 3-6 nm. The length of DNA per base is about 0.34 nm and the length of 12 bp DNA is about 4 nm.

(3) Fig. 2i and others: I find the nomenclature MLDS-N (etc) confusing. To me it denotes the protein minus the amino terminus. Consider using superscript instead throughout?

Response: We thank the reviewer's useful advice. We replaced throughout style with superscript style in the revised manuscript.

(4) Fig 2h: some cartoons like this one are confusing - it looks as if both samples are mixed before centrifuging. Is this what is meant?

Response: We thank the reviewer's question. We corrected them in the revised manuscript.

(5) Fig. 2i and others: Are we comparing equal % of adiposomes and solution? This is not clear from methods or legend.

Response: We thank the reviewer's useful comment. Yes, we are comparing equal volume of adiposome and solution fractions. We did mention it in Methods: "The samples were adjusted to equal volumes."

(6) Fig 3f: why doesn't the KO+MLDS FULLY rescue? You are putting back exactly what was missing. This needs an explanation.

Response: We thank the reviewer's critical question. In fact, we do not know the answer. Our explanation is that since MLDS deletion also enlarges LD size (Ding, Y. et al. *J Lipid Res* 53, 399-411 (2012)), it may be a multifunctional protein. Therefore, deletion of MLDS may change bacteria dramatically and re-expression may not be able to 100% recover the growth phenotype.

(7) Fig. 3j: is it fair to compare the tail moments if the signal (fluorescence brightness) is so different? Could not some of the measurement be affected by detection sensitivity?

Response: We thank the reviewer for this comment. The neutral bacterial comet assay was carried out by combining previous methods (the reference numbers are 28, 62, and 63 in the revised manuscript). The WT and MLDS KO cells were treated under the same conditions, including lysis, DNA unwinding, electrophoresis, and staining. And the intensity of laser and exposure time for the observation were also same. We quantified the tail moments in 50 independent cells and the conclusion was raised from the quantification result of the experiment in Fig. 3j (bottom). The images in Fig. 3j (top) were selected for representing the difference between WT and MLDS KO cells.

(8) I'd like to see how much MLDS is in the droplet vs. cytosol fractions.

Response: We thank the reviewer's useful comment. The original figure shows that based on equal protein load, MLDS is highly enriched in LD fraction. In fact, equal volume load is a proper method to determine the distribution of a protein in different cellular fractions. During cell fractionation LDs are lost dramatically while other cellular fractions are not, it is very hard to make an equal volume load.

(9) Related to above, Fig. 4b: If equal protein are loaded, one gets a false idea of distribution, as droplets have so little protein. The cytosol may have 50% of ro02105, for example, while the specific activity is very much higher in the droplet fraction.

Response: We thank the reviewer's critical comments and understand the reviewer's concern. As we mentioned in the previous response, although it is not rigorous to compare the distribution between LD fraction and other fractions by equal protein load, we think that the equal protein load is better way to determine whether one protein is LD-associated protein. We judge whether one protein is LD-associated protein by checking if the protein is rich in LD fraction and by comparing with LD protein (Ro05869), non-LD protein (GFP) and dual-localization protein (Ro05469). Our conclusion from Fig. 4b is that RHA1_ro02105 was a LD-associated protein.

(10) Fig 5a: This is most confusing! There are 2 ~40-bp fragments shown in the bottom of the panel. But only one is run out in 5b, right. I'm confused about these two small fragments.

Response: We thank the reviewer's question and we are sorry for the confusion. Fig. 5a shows the DNA motifs for MLDSR binding. The 103 bp DNA is the upstream sequence of the genes *mldsr* and *mlds* in the genome. Only one 43 bp DNA fragment was shown in Fig. 5a and it was narrowed down from the 103 bp DNA using EMSAs (Fig. 5b and SI Fig. 6a-f). The EMSAs using 103 bp DNA and 43 bp DNA are shown in Fig. 5b.

(11) Fig 5d and 5e: why wasn't Motif 2 by itself analyzed?

Response: We thank the reviewer's question and we are sorry for the confusion. In fact, Motif 2 was analyzed in Fig. 5f.

(12) Text regarding Fig 3c: is it far to score proximity in extremely low nitrogen as the cells are so much shorter? The DNA has no where to go other than by a droplet.

Response: We thank the reviewer's critical comments. The reviewer is right that the cells became shorter in 0.1 g/L nitrogen culture. Even though the cells are shorter in

extremely low nitrogen, there is still adequate space in bacterial cells as showing in following figure. Thus, we can still distinguish cytosolic and LD-associated genomic DNA.

(13) lines 221-222: the palindrome is not obvious from this sequence; please make this clearer

Response: We thank the reviewer's useful comment. We corrected it "define a 16 bp palindromic sequence, 5'-GNT (A/T) GCTNNTGCTANC-3'" and the symmetry axis is between the middle "NN". Thus, the palindrome is obvious.

(14) line 257: that alpha2 shares the two functions (droplet localization and DNA binding) is an overinterpretation. That sequence may be required for a general folding function of the protein rather than the more specific functions discussed.

Response: We thank the reviewer's critical comments. In our results, the DNA binding domain of MLDSR was helices $\alpha 2$ - $\alpha 4$ (25-78 amino acids) (Fig. 6c-e). The predicted architecture domain analysis of MLDSR shows that its N-terminal sequence (29-84 amino acids) is xenobiotic response element (XRE) helix-turn-helix (HTH) DNA-binding motif (SI Fig. 5a), which shows that our results match the predicted analysis. And the α -helix in HTH domain (that is $\alpha 2$) is predicted to bind DNA similar to lambda repressor (SI Fig. 7m). Thus it suggests that $\alpha 2$ should not be for a general folding function of the protein. In addition, we also think that the LD targeting domain and DNA binding domain of MLDSR is not only $\alpha 2$. Based on the results, the LD targeting domain is $\alpha 2$ - $\alpha 6$ and the DNA binding domain is $\alpha 2$ - $\alpha 4$, which suggests that "the LD localization and DNA binding regions of MLDSR share at least one α -helix, $\alpha 2$ ". Thus, we think that the statement represents the results.

(15) line 353: confusing: "composition of the membrane layer surrounding them" as in some cases there is an absence of a membrane.

Response: We thank the reviewer's useful comment. It is true that in some cases there is an absence of a membrane and these BMCs without membranes are named after protein-based BMCs. The reference numbers are 46 and 47 in the manuscript.

Cornejo, E., Abreu, N. & Komeili, A. Compartmentalization and organelle formation in bacteria. *Curr Opin Cell Biol* 26, 132-8 (2014).

47. Shively, J.M. Complex intracellular structures in prokaryotes (Springer-Verlag, Berlin Heidelberg, Germany, 2006).

(16) General comment: The DNA is not tightly adhering to droplets in the micrographs, but is spread out such that much of the DNA is 1-2 micrometers away. Why is there such a large effect on DNA stability, especially after UV damage? Do the authors believe that droplets can stabilize DNA even when the DNA is not that close to the droplets?

Response: We thank the reviewer's critical comments. Firstly, based on our study, DNA did not directly contact with LD neutral lipids (Fig. 2i and 2j, lane 2). Protein MLDS was the link to connect them. Thus it is understandable that DNA is not very close to LDs (neutral lipids) in SIM images. Secondly, according to the results (SI Fig. 4b-d), we speculate that LDs can stabilize genomic DNA by involving in the DNA repair process of the bacterium via the nucleotide excision repair (NER) system. Thirdly, we speculate that more genomic DNA locate on LDs after UV exposure and that DNA repair proteins (like UvrA) on LDs are more efficient for repairing DNA damage.

Reviewer #4 (Remarks to the Author):

The authors of the manuscript entitled "Lipid Droplets Mediate Genomic DNA Localization and Transcription to Promote DNA Stability and Bacterial Survival in Extreme Environments" studied the protective role of lipid droplets to genomic DNA. The authors concluded that MLDS protein and MLDSR are involved in the process by binding DNA and proving a controlling method for the lipid droplets.

Although the study has merit and seems to have been competently carried out, the authors never discuss aspects that may have greatly influenced the binding of DNA to the lipid droplets such as the lipidic nature of the LD, in particular the phospholipids in the monolayer membrane that will contact with the DNA, and the inherent polarity of the DNA chain. Both electrostatic and hydrophobic/hydrophilic interactions should have been considered in the interpretation of the results. The authors may decide that they only wanted to study the role of proteins in LD but the lipid character of LD must play a decisive role in the binding of DNA and other molecules. The authors

should at least discuss this aspect in the discussion and also the chemical properties of both lipids and proteins in LB, and DNA.

Response: We thank the reviewer's critical comments. We included several sentences to discuss the relationship between monolayer phospholipid membrane of LDs and DNA in the Discussion of the revised manuscript. "Our current study presents that LDs recruit and protect genomic DNA through MLDS. In fact, proteins are necessary for DNA association to bacterial LDs. LDs are coated by phospholipid monolayer membrane and phospholipids are negatively charged (phosphatidic acid, phosphatidyl inositol and phosphatidyl serine) or electrically neutral (phosphatidyl ethanolamine, phosphatidyl choline, and sphingomyelin). The DNA chain is negatively charged and thus there should not be electrostatic interaction between DNA and phospholipid membrane. Our *in vitro* experiments show that LD mimics adiposomes could not bind DNA without specific protein MLDS (Fig. 2i and 2j, lane 2), suggesting that there is no other detectable binding including hydrophobic or hydrophilic interaction between LD phospholipids and DNA." Thus, we think that proteins are key bridge that connects LDs and DNA. LD-associated proteins are specific to interact with LDs. In addition, we guess that the specific chemical and physical properties of the LD membrane may contribute to the specificity and efficiency with which LDs participate in certain cell processes. For now, we have known that some LD-associated proteins contain specific structures. For example, ACSL3, GPAT4, and DGAT2 contain hydrophobic hairpin/helix, and PLINs, and Cidea contain amphiphilic helices (Kory, N. Farese, R. V. Walther, T. C. Targeting Fat: Mechanisms of Protein Localization to Lipid Droplets. Trends in Cell Biology, 2016). MLDS and MLDSR in the study also contain amphiphilic helices domain. Above all, LDs can specifically bind proteins and some LD-associated proteins bind DNA such as MLDS. LDs can also change or occupy the proteins to affect their ability of binding DNA, including MLDSR and JLP.

The association of genomic DNA with LDs is not clear by the SIM and confocal microscopy images since what the white arrows are pointing to varies in the several images. Besides, the name of the "green" dye should be Hoechst® and not Hoechst. However, according to the manufacturer of this dye, the DNA should be stained blue. Green emission has been observed when the dye suffers photobleaching as described by Żurek-Biesiada et al. J Fluoresc. 2014; 24(6): 1791–1801. What is the explanation for the green colour e.g. in SFig 1?

Response: We thank the reviewer's critical comments. We are sorry for the confusion. Based on our study, DNA did not directly contact with LD neutral lipids specifically (Fig. 2i and 2j, lane 2). In between, there are phospholipid membrane and proteins. Protein MLDS was the link to connect them. Thus, there is no overlapping signal between DNA and LD signal in SIM images. What the white arrows are pointing to in images (Fig. 1c, 3c, 7a, and SFig. 1b) indicates the association between LDs and

DNA. Since DNA can co-localize with LD marker protein such as Ro05869-GFP in co-localization assay, there was overlapping between DNA and Ro05869-GFP signal in confocal microscopy images. What the white arrows are pointing to in images (Fig. 1d, 3d, and 7b) indicates the overlapping between LD marker protein and DNA. Furthermore, we corrected Hoechst word in the revised manuscript and thank the reviewer again for informing us. In addition, we are sorry for the confusion about SFig. 1b. In SFig. 1b, Hoechst signal was excited with 405 nm laser and the DNA was stained blue. But the intensity of blue is too weak and it's hard to observe whether the blue associate with the red of LD signal, we replaced the blue with the green by software without any other change to observe the association between the green of DNA and the red of LDs. Above all, the original color of DNA in SFig. 1b is blue, and the blue then is replaced with the green by software for observation.

The work is also very dependent on microscopy images. For example, L492 “Fluorescence intensity from SYTOX staining and overexpressed GFP-tagged LD markers was quantified with Image J software.” However, intensity measurements in images are not a “bullet proof” technique since the amount of light reaching the sample on each image may vary. The authors should have measured the intensity peaks e.g. by fluorometry or flow cytometry or present the results as intensity ratios.

Response: We thank the reviewer's critical comments. The co-localization analysis was modified from the method described in the paper (Bei-Bei Chu et. al, Cholesterol Transport through Lysosome-Peroxisome Membrane Contacts. Cell, Volume 161, Issue 2, 2015, Pages 291-306). In the assay, co-localization was evaluated using the JACoP plugin of Image J, by calculating Mander's fraction of the A image overlapping the B image, which represented the fraction of DNA co-localizing with a LD marker. We make sure to use the consistent intensity in different samples as much as possible. We added the intensity thresholds used in each channel for this analysis in the revised Methods although we did not measure the intensity peaks. We think that the represented information is sufficient for the quantification.

Other comments (L=Line):

- L8: “through the major LD protein, microorganism lipid droplet small (MLDS), which (...)”

The phrase should be “through the major LD protein, microorganism lipid droplet small (MLDS) protein, which”. In fact, “microorganism lipid droplet small” does not make much sense in many of the phrases in the text. This also appears in L35.

Response: We thank the reviewer's useful suggestion. We corrected them in the revised manuscript.

L30: “*Rhodococcus opacus* PD630 (PD630) and *Rhodococcus jostii* RHA1 (RHA1) are known to contain more LDs and TAG content than any other bacteria identified to date.”

This was true in 2001 but is it still?

Response: We thank the reviewer’s useful comment and reasonable argument. We changed the sentence and added the recent reference in the revised manuscript as “*Rhodococcus opacus* PD630 (PD630) and *Rhodococcus jostii* RHA1 (RHA1) are oleaginous model bacteria that are known to contain large amount of LDs and TAG content.”

L84: “We then utilize” should be “We then utilized”

Response: We thank the reviewer’s useful suggestion. We corrected it in the revised manuscript.

L115: “DNA and proteins were then incubated with the adiposomes, and then the reaction was centrifuged to separate the adiposomes from the reaction solution (Fig. 2h).”

To which reaction are the authors referring to? DNA and protein incorporation in LB should not be the result of a reaction.

Response: We thank the reviewer’s useful comment. We replaced “reaction” with “mixture” in the revised manuscript.

L509: “The cells were treated with UV exposure” should be “The cells were exposed to UV light”.

Besides, the authors refer that the cells grew to OD~2 in liquid medium and were then cultured on LB solid medium. No information on the age of cells (stationary or exponential phase), initial number of cells used to inoculate the solid medium, and time during which the cells were cultured before being is exposed to UV light is referred. It is known that number of cells and age of culture influence greatly the survival to UV light.

Response: We thank the reviewer’s useful comments. We changed the sentence to “The cells were exposed to UV light” and added the information in the revised manuscript. The initial number of cells was ~1000 (Fig. 3h, 0 J/m² UV exposure) and we utilized bacteria under late exponential phase to perform the experiment because LDs are sufficient under late exponential phase, and it is consistent with other experiments in the study.

L544: How were the cells homogenized? The method may influence the amount and quality of LD recovered.

Response: We thank the reviewer's useful comment. We followed the previous study (the reference number is 64). The resuspended cells were homogenized by passing through a French pressure cell three times at 100 MPa, 4°C. We added it in the revised manuscript.

L547: "To prepare the LD protein sample, 1 mL of chloroform:acetone (1:1, v/v) was added. LD proteins were mixed with 2×SDS sample buffer and denatured at 95°C for 5 min."

Were the proteins from the LD i) extracted and used whilst others were denatured or ii) extracted and denatured?

Response: We thank the reviewer's useful question. We are sorry for the missing information. In the revised manuscript, we changed the original sentence to "To prepare the LD protein sample, 1 mL of chloroform:acetone (1:1, v/v) was added. LD proteins were extracted, dissolved in 2×SDS sample buffer, and denatured at 95°C for 5 min."

Thank you for your consideration of this revised manuscript.

Reviewers' comments:

Reviewer #1 (Remarks to the Author):

The authors have addressed my previous comments

Reviewer #2 (Remarks to the Author):

The authors have addressed most of my comments in the rebuttal letter. However, I feel they have not revised to manuscript at corresponding positions to clarify the confusion any better. Please find my responses to the places that require revision in the manuscript. The original comments and responses from the authors are copied for better reference, whereas my additional comments are in bold.

1. The association between genomic DNA and LDs under various conditions is one important piece of data supporting the model. Therefore I think it's necessary to clarify the imaging and quantification methods from the following few aspects:

1) The authors used two pairs of labeling schemes and microscopies to study the association: (i) staining DNA and LD lipid (with LipidTox) and imaging with SIM, and (ii) staining DNA and LD marker Ro0568 (fused with GFP) and imaging with confocal. It is unclear why colocalization analysis can only be performed in case (ii). Based on SI Fig 1c, Ro0568 stained mostly outside the LipidTox staining region, which seems to be the reason for being used for colocalization analysis, as one can see overlapping DNA and Ro0568 signal, whereas the for case (i) it's not very obvious. Nevertheless, the case (i) is still used for many comparisons, in which arrows are used to indicate association. What is the quantitative criterion to be determined as association? Can you quantify this, and compare with the colocalization analysis?

Response: We thank the reviewer's critical comments. LDs are spherical structure with neutral lipid core covered by phospholipid membrane and proteins. Based on our study, DNA did not directly contact with LD phospholipids and neutral lipids (Fig. 2i and 2j, lane 2). In between DNA and LD neutral lipids there were phospholipid membrane and proteins. In our finding, protein MLDS was the link to connect LDs and DNA. Thus, no overlapping signal between DNA and LD neutral lipids (LipidTox) should be detected. That is why it is hard to quantify the overlapping between DNA and LDs (neutral lipid staining) in SIM images. Using double staining of DNA and neutral lipids, we could only compare the proximity of DNA and LD neutral lipids between MLDS WT and MLDS KO cells. To verify the finding of double staining experiment, we performed the colocalization assay. In the study, we used the SIM observation in three places mainly (Fig. 1c, 3c and 7a). When we wanted to compare the association of DNA and LDs under various conditions quantitatively, we always performed the colocalization assays (Fig. 1d-e, 3d-e, and 7b-c). Since LDs in bacteria are very small and average size is about 0.3 μm diameter, visualizing bacterial LDs is difficult. Two experimental designs can be complementary advantage as well as verify each other. Furthermore, on top of imaging the results from biochemical and molecular experiments including *in vitro* assays sufficiently support our conclusion.

2) Related to SI Fig 1c, it seems that the colocalization (the overlapping signal) between Ro0568 and LipidTox staining is weaker in MLDS KO case compared to WT. Does that mean the association of the LD marker to LDs is also regulated in some way, which would affect the quantification of DNA/LD colocalization if used as a reference?

Response: We thank the reviewer's critical comments. We are sorry for the confusion. LDs are spherical structure with neutral lipid core and phospholipid membrane. In general, LD proteins do not directly contact with LD neutral lipids. Therefore, the overlapping signal between proteins and LD neutral lipids should not be obvious. In current study, we did not try to use overlapping signal to identify the association of proteins to LDs. In 2D image, **LDs were red round dots with LipidTox staining while the Ro05869-GFP was "ring" structure, which is in agreement with LD physiology.** In addition, according to the results (SI Fig. 1d), Ro05869 expression and the protein on LDs were similar in WT and MLDS KO cells, and almost all the signals detected by anti-GFP were in LD fractions. The experiment in SI Fig. 1c was designed to identify that Ro05689 is a LD-associated protein and the weaker signals in MLDS KO cells (the Ro05689-GFP signal and lipidTox signal were both weaker) may be due to the weaker intensity of lasers. In colocalization assay, the almost same intensity of lasers was used and the green signal of Ro05869-GFP was similar in WT and MLDS KO cells (Fig. 1d). Above all, we think that Ro05869 can be a specific LD marker for colocalization assay.

I am convinced by the response, but I think it's better for the author to put some of the explanation in the text directly, which I think will make it easier to understand. For example, the expected "ring-like" structure of Ro05689-GFP when first introduced in the text.

3) Need more information about the colocalization analysis in the method. For example, whether this is done in 2D or 3D, and what are the intensity thresholds used in each channel for this analysis? These parameters shall affect the quantification result.

Response: We thank the reviewer's useful suggestion. We have added the information in the method in the revised manuscript. They are 2D images. In WT cells cultured in MSM with 0.5 g/L NH₄Cl for control, it was used the mean threshold of 1,440 for SYTOX signal and the mean threshold of 670 for RHA1_ro05869-GFP signal. In MLDS KO cells, it was used the mean threshold of 1,330 for SYTOX signal and the mean threshold of 610 for RHA1_ro05869-GFP signal. In MLDSR KO cells, it was used the mean threshold of 1,150 for SYTOX signal and the mean threshold of 800 for RHA1_ro05869-GFP signal. In WT cells cultured in MSM with 0.1 g/L NH₄Cl, it was used the mean threshold of 1,400 for SYTOX signal and the mean threshold of 760 for RHA1_ro05869-GFP signal. In MLDSR OE cells, it was used the mean threshold of 1,730 for SYTOX signal and the mean threshold of 1,150 for MLDSR-GFP signal. The colocalization analysis was modified from the method described in the paper (Bei-Bei Chu et. al, Cholesterol Transport through Lysosome-Peroxisome Membrane Contacts. Cell, Volume 161, Issue 2, 2015, Pages 291-306).

Please clarify in the methods for quantification in 2D images, whether they are the middle z stack or the 2D projection of a 3D images. In addition, threshold are slightly different

between different conditions for colocalization analysis. Why not use the same threshold, and how were these threshold chosen?

3. The paragraph between lines 125-131 is very confusing to read. The interpretation does not strongly reflect SI fig 2d. (1) Authors interpreted the data as “both fusion proteins could bind DNA to the adiposome”, however, in the lane for condition “MLDS-N-H1 + DNA + adiposome” (3&7), majority of DNA is in the solution. (2) It’s also unclear to me why SI fig 2d can lead to the conclusion of “LD-targeting and DNA-binding domains of MLDS were at its N-terminus and C-terminus respectively”, because there is no control to show H1 without fusing to MLDS-N cannot be targeted to adiposome. In fact, this is possible because ADRP seems to be targeted to adiposome without any fusion. I think the most straightforward evidence to support this conclusion is to do the same assay in the MLSD-C (without the N terminal targeting sequence) and demonstrate that MLSD-C binds to DNA but is not in adiposome. (3) What are the three bands in Lane 3?

Response: We thank the reviewer’s critical comments.

(1) It is true that majority of DNA is in the solution. But we can judge whether a protein binds DNA to adiposomes or not through 1) comparing the amount of DNA in adiposome fraction and solution fraction, 2) comparing the amount of DNA in adiposome fraction between target protein and control protein, and 3) checking whether DNA is decreased in solution fraction compared with original DNA (lane 1). According the result, the DNA of lane 3 was less than that of lane 7, but more than that of lane 2, and the DNA of lane 7 was less than that of lane 1. Thus, MLDS-N-H1 was able to bind DNA to adiposomes.

In order to better present anything related to quantity comparison, I suggest the authors make a bar graph to directly compare the relevant lanes in the gel.

(2) We are sorry for the confusion. It is reported that histones cannot locate on LDs by themselves, an anchor protein Jabba is necessary for the location (Li, Z. et al. Lipid droplets control the maternal histone supply of Drosophila embryos. *Curr Biol* 22, 2104-13 (2012)). We mentioned the information in introduction in the manuscript, “Furthermore, in another study, histones were found localized to LDs via the anchor protein Jabba in Drosophila”. ADRP is a LD resident protein in mammalian cells (we included the information in the revised manuscript). Thus, ADRP can target to adiposomes (Wang, Y. et al. Construction of Nano-Droplet/Adiposome and Artificial Lipid Droplets. *ACS Nano* (2016)) and histone H1 could not. Therefore, the targeting of MLDS-N-H1 to adiposomes suggests that MLDS-N is the targeting domain. In fact, our previous work shows that MLDS-N contains the LD-targeting domain (Ding, Y. et al. Identification of the major functional proteins of prokaryotic lipid droplets. *J Lipid Res* 53, 399-411 (2012)). In current study, MLDS-N was found to be localized to adiposomes (Fig. 2i, lane 3 and 4). The result in SI Fig. 2d further supports the

conclusion. The conclusion that MLDS-C is the DNA binding domain has been represented in the study (Fig. 2). The result in SI Fig. 2d further supports the conclusion.

The authors should make revision in the main text to clarify the confusion. The paragraph that was unclear to me remain unchanged at all in the revised manuscript. I think the highlighted part should be briefly reflected in the related paragraph to improve the reasoning that lead to the conclusion.

(3) We guess that because the fusion protein is instable. Full-length MLDS-N-H1 is degraded into several pieces. Based on the prediction of molecular weight, the highest band of the three bands is putative full-length MLDS-N-H1.

4. The authors concluded that the mRNA and protein levels of MLDS were significantly reduced in both the deletion and overexpression mutants, whereas in Fig 4f, it appears to me that KO and OE cases have less proteins in general, as evidenced by the bands on the top, especially as position of 95kDa and 55kDa. I think the authors should have a reference band for rigorous comparison.

Response: We thank the reviewer's critical comments. In fact, it is hard to find a robust reference band in bacterial LD fraction. Though Ro05869 is considered, we think that comparing the total proteins of LD fraction in general is better. In figure 4f, it is true that the proteins of KO lane were less than that of WT lane. But the decrease of MLDS was more significant and we repeated the experiment in SI F5d. The proteins of KO lane and WT lane were similar, and MLDS was also decreased dramatically. In figure 4f, the proteins of OE lane were similar with that of WT lane because of the large amount of MLDSR-GFP. Thus, protein levels of MLDS were significantly reduced in both the deletion and overexpression mutants.

I am not entirely convinced by comparing with the total protein level in each case. I suggest use some house-keeping protein as a normalization to quantity comparison, and put the quantification in the bar graph instead of having the readers judge by eyes.

Minor:

Rate constant should be lower "*k*" instead of "K" in some of the figures.

Reviewer #3 (Remarks to the Author):

This version is much improved, and I have no new critical comments. While I'm satisfied with most of my original issues, four of them remain and I ask the authors to again address them; otherwise, I'm fairly certain these point will confuse the readership. Numbers below correspond to my original critical comments. (I'm Reviewer 3.)

(2) figure 2d: a major gel shift occurs when the protein/DNA ratio reaches about 50. The authors term this (in their rebuttal letter) a "confused image." As this still indicates a major binding shift to me, the authors just need to address the point in the text or figure legend. If this gel behavior is an artifact at this ratio, this just needs to be stated as such.

(6) The fact that the KO+MLDS doesn't fully rescue the KO phenotype just should be mentioned in the text, even if the explanation is not known. To me the omission is glaring.

(7) figure 3g and 3j: Please just use images where the intensity of the main spot between WT and KO samples is similar, to allow the reader to directly compare the comet morphology.

(10) In figure 5a, there are two 43 bp sequences at the bottom of the panel. It still is not clear to me which one of these is used in panel b (right). I didn't understand the explanation of the authors.

Reviewer #4 (Remarks to the Author):

The authors of the manuscript entitled "Lipid Droplets Mediate Genomic DNA Localization and Transcription to Promote DNA Stability and Bacterial Survival in Extreme Environments" responded satisfactorily to the majority of my questions. I would have preferred if the authors had presented quantitative information gathered by techniques such as fluorometry, flow cytometry or real image analysis (as intensity ratios of 2 dyes present in the same image and not of measurements made with ImageJ) but I accept these results. However, I disagree with the explanation regarding electrostatic and hydrophobic interactions between the phospholipids monolayer of lipid droplets and DNA.

The authors claim that "LDs are coated by phospholipid monolayer membrane and phospholipids are negatively charged (phosphatidic acid, phosphatidyl inositol and phosphatidyl serine) or electrically neutral (phosphatidyl ethanolamine, phosphatidyl choline, and sphingomyelin). The DNA chain is negatively charged and thus there should not be electrostatic interaction between DNA and phospholipids membrane."

Cellular membranes may contain zwitterionic phosphatidylcholines (PC) and phosphatidylethanolamine (PE) as well as anionic phosphatidylinositol (PI) and phosphatidylserine (PS). The phospholipid monolayer of the lipid droplets is complex, resulting from electrostatic and hydrophobic interactions between different phospholipids which have impact on lipid packing. The polyanionic DNA molecule interacts with the phospholipid headgroups and it's repulsed by negative charges present in the phospholipids. However, the presence of counterions such as Ca ions may trigger electrostatic attraction (vide doi:10.1016/j.cis.2014.01.016; DOI: 10.1039/c3sm51419f; <http://dx.doi.org/10.1063/1.1325230>, doi: <http://dx.doi.org/10.1063/1.1436077>).

It is also known that lipid droplets in actinomycetes have a lipid core (with e.g. triacylglycerols and wax esters) covered by a monolayer of phospholipids, which prevents coalescence or denaturation of cytoplasmic proteins due to hydrophobic interactions (Hanish et al. 2006; doi: 10.1128/AEM.00584-06). Other studies have shown that most proteins that bind to the phospholipids monolayer of lipid droplets do it via hydrophobic interactions (e.g. Kory et al. 2016;

<http://dx.doi.org/10.1016/j.tcb.2016.02.007>). Attractive interactions between the nitrogen bases of DNA and hydrophobic surfaces have been demonstrated by Cardenas et al. 2003 (DOI: 10.1021/la026747f).

The authors should read carefully the literature to assess the phospholipid composition of lipid droplets in *Rhodococcus jostii* RHA1 and discuss the electrostatic and hydrophobic interactions between those lipids and DNA and also between lipids and protein MLDS as its position depends on the same forces.

Reviewer #1 (Remarks to the Author):

The authors have addressed my previous comments

Response: We thank the reviewer's previous critical comments and suggestions again. These comments and suggestions are indeed very helpful. We also appreciate the reviewer's satisfaction for our responses.

Reviewer #2 (Remarks to the Author):

The authors have addressed most of my comments in the rebuttal letter. However, I feel they have not revised to manuscript at corresponding positions to clarify the confusion any better. Please find my responses to the places that require revision in the manuscript. The original comments and responses from the authors are copied for better reference, whereas my additional comments are in bold.

1. The association between genomic DNA and LDs under various conditions is one important piece of data supporting the model. Therefore I think it's necessary to clarify the imaging and quantification methods from the following few aspects:

1) The authors used two pairs of labeling schemes and microscopies to study the association: (i) staining DNA and LD lipid (with LipidTOX) and imaging with SIM, and (ii) staining DNA and LD marker Ro0568 (fused with GFP) and imaging with confocal. It is unclear why colocalization analysis can only be performed in case (ii). Based on SI Fig 1c, Ro0568 stained mostly outside the LipidTOX staining region, which seems to be the reason for being used for colocalization analysis, as one can see overlapping DNA and Ro0568 signal, whereas the for case (i) it's not very obvious. Nevertheless, the case (i) is still used for many comparisons, in which arrows are used to indicate association. What is the quantitative criterion to be determined as association? Can you quantify this, and compare with the colocalization analysis?

Response: We thank the reviewer's critical comments. LDs are spherical structure with neutral lipid core covered by phospholipid membrane and proteins. Based on our study, DNA did not directly contact with LD phospholipids and neutral lipids (Fig. 2i and 2j, lane 2). In between DNA and LD neutral lipids there were phospholipid membrane and proteins. In our finding, protein MLDS was DNA anchor on LD phospholipid membrane. Thus, no overlapping signal between DNA and LD neutral lipids (LipidTOX) should be detected. Therefore, it is hard to quantify the overlapping between DNA and LDs (neutral lipid staining) in SIM images. Using double staining of DNA and neutral lipids, we could only compare the proximity of DNA and LD neutral lipids between MLDS WT and MLDS KO cells. To verify the finding of double staining experiment, we performed the colocalization assay. In the study, we

used the SIM observation in three places mainly (Fig. 1c, 3c and 7a). When we wanted to compare the association of DNA and LDs under various conditions quantitatively, we always performed the colocalization assays (Fig. 1d-e, 3d-e, and 7b-c). Since LDs in bacteria are very small and average size is about 0.3 μm diameter, visualizing bacterial LDs is difficult. Two experimental designs can be complementary advantage as well as verify each other. Furthermore, on top of imaging the results from biochemical and molecular experiments including in vitro assays sufficiently support our conclusion.

2) Related to SI Fig 1c, it seems that the colocalization (the overlapping signal) between Ro0568 and LipidTOX staining is weaker in MLDS KO case compared to WT. Does that mean the association of the LD marker to LDs is also regulated in some way, which would affect the quantification of DNA/LD colocalization if used as a reference?

Response: We thank the reviewer's critical comments. We are sorry for the confusion. LDs are spherical structure with neutral lipid core and phospholipid membrane. In general, LD proteins do not directly contact with LD neutral lipids **except a few cases**. Therefore, the overlapping signal between proteins and LD neutral lipids should not be obvious. In current study, we did not try to use overlapping signal to identify the association of proteins to LDs. In 2D image, LDs were red round dots with LipidTOX staining while the Ro05869-GFP was "ring" structure, which is in agreement with LD physiology. In addition, according to the results (SI Fig. 1d), Ro05869 expression and the protein on LDs were similar in WT and MLDS KO cells, and almost all the signals detected by anti-GFP were in LD fractions. The experiment in SI Fig. 1c was designed to identify that Ro05689 is a LD-associated protein and the weaker signals in MLDS KO cells (the Ro05689-GFP signal and LipidTOX signal were both weaker) may be due to the weaker intensity of lasers. In colocalization assay, the almost same intensity of lasers was used and the green signal of Ro05869-GFP was similar in WT and MLDS KO cells (Fig. 1d).

Above all, we think that Ro05869 can be a specific LD marker for colocalization assay.

I am convinced by the response, but I think it's better for the author to put some of the explanation in the text directly, which I think will make it easier to understand. For example, the expected "ring-like" structure of Ro05689-GFP when first introduced in the text.

Response: We thank the reviewer's suggestion. We added the information about the "ring-like" structure of Ro05689-GFP in the legends of Figure 1d and Supplementary Figure 1c in the revised manuscript.

3) Need more information about the colocalization analysis in the method. For example, whether this is done in 2D or 3D, and what are the intensity thresholds used in each channel for this analysis? These parameters shall affect the quantification result.

Response: We thank the reviewer's useful suggestions. We added the information in the method of the revised manuscript. They are 2D images. In WT cells cultured in MSM with 0.5 g/L NH₄Cl for control, it was used the mean threshold of 1,440 for SYTOX signal and the mean threshold of 670 for RHA1_ro05869-GFP signal. In MLDS KO cells, it was used the mean threshold of 1,330 for SYTOX signal and the mean threshold of 610 for RHA1_ro05869-GFP signal. In MLDSR KO cells, it was used the mean threshold of 1,150 for SYTOX signal and the mean threshold of 800 for RHA1_ro05869-GFP signal. In WT cells cultured in MSM with 0.1 g/L NH₄Cl, it was used the mean threshold of 1,400 for SYTOX signal and the mean threshold of 760 for RHA1_ro05869-GFP signal. In MLDSR OE cells, it was used the mean threshold of 1,730 for SYTOX signal and the mean threshold of 1,150 for MLDSR-GFP signal. The colocalization analysis was modified from the method described in the paper (Bei-Bei Chu et. al, Cholesterol Transport through Lysosome-Peroxisome Membrane Contacts. Cell, Volume 161, Issue 2, 2015, Pages 291-306).

Please clarify in the methods for quantification in 2D images, whether they are the middle z stack or the 2D projection of a 3D images. In addition, threshold are slightly different between different conditions for colocalization analysis. Why not use the same threshold, and how were these threshold chosen?

Response: We thank the reviewer's critical comments and questions. These 2D images are the middle z stack and we added the information in the method of the revised manuscript.

We slightly changed the threshold for: 1) reducing the background of fluorescence in the images, and 2) decreasing the overexposure of fluorescence in the images. Through changing the threshold, we avoided high background and overexposure of fluorescence that can affect the colocalization analysis. We present here one example.

Appropriate setting

High background

Overexposure

There are three groups of images (A/a, B/b, and C/c). A1-A3 are real fluorescent images and a1-a3 represent the background (blue) and fluorescent intensity (red) of

A1-A3, respectively. The groups of B/b and C/c are same. By comparing with the groups of B/b and C/c, the group of A/a is accepted.

Even though we tried our best to use similar parameters to take images, there still were some differences in the experiments. Therefore, through monitoring the distribution of blue and red in the lower panel, we can select the appropriate setting (no high background and overexposure) by changing the threshold and we can also make these images for quantification consistent. That's why these thresholds are slightly different sometimes.

3. The paragraph between lines 125-131 is very confusing to read. The interpretation does not strongly reflect SI fig 2d. (1) Authors interpreted the data as “both fusion proteins could bind DNA to the adiposome”, however, in the lane for condition “MLDS-N-H1 + DNA + adiposome” (3&7), majority of DNA is in the solution. (2) It's also unclear to me why SI fig 2d can lead to the conclusion of “LD-targeting and DNA-binding domains of MLDS were at its N-terminus and C-terminus respectively”, because there is no control to show H1 without fusing to MLDSN cannot be targeted to adiposome. In fact, this is possible because ADRP seems to be targeted to adiposome without any fusion. I think the most straightforward evidence to support this conclusion is to do the same assay in the MLSD-C (without the N terminal targeting sequence) and demonstrate that MLSD-C binds to DNA but is not in adiposome. (3) What are the three bands in Lane 3?

Response: We thank the reviewer's critical comments.

(1) It is true that majority of DNA is in the solution. But we can judge whether a protein binds DNA to adiposomes or not through 1) comparing the amount of DNA in adiposome fraction and solution fraction, 2) comparing the amount of DNA in adiposome fraction between target protein and control protein, and 3) checking whether DNA is decreased in solution fraction compared with original DNA (lane 1). According the result, the DNA of lane 3 was less than that of lane 7, but more than that of lane 2, and the DNA of lane 7 was less than that of lane 1. Thus, MLDS-N-H1 was able to bind DNA to adiposomes.

In order to better present anything related to quantity comparison, I suggest the authors make a bar graph to directly compare the relevant lanes in the gel.

Response: We thank the reviewer's suggestion. We added the bar graph by quantifying the relative DNA content in lanes using Image J software in SI fig. 2e in the revised manuscript.

(2) We are sorry for the confusion. It is reported that histones cannot locate on LDs by themselves, an anchor protein Jabba is necessary for the location (Li, Z. et al. Lipid droplets control the maternal histone supply of *Drosophila* embryos. *Curr Biol* 22, 2104-13 (2012)). We mentioned the information in introduction in the manuscript, “Furthermore, in another study, histones were found localized to LDs via the anchor protein Jabba in *Drosophila*”. ADRP is a LD resident protein in mammalian cells (we included the information in the revised manuscript). Thus, ADRP can target to adiposomes (Wang, Y. et al. Construction of Nano-Droplet/Adiposome and Artificial Lipid Droplets. *ACS Nano* (2016)) and histone H1 could not. Therefore, the targeting of MLDS-N-H1 to adiposomes suggests that MLDS-N is the targeting domain. In fact, our previous work shows that MLDS-N contains the LD-targeting domain (Ding, Y. et al. Identification of the major functional proteins of prokaryotic lipid droplets. *J Lipid Res* 53, 399-411 (2012)). In current study, MLDS-N was found to be localized to adiposomes (Fig. 2i, lane 3 and 4). The result in SI Fig. 2d further supports the conclusion. The conclusion that MLDS-C is the DNA binding domain has been represented in the study (Fig. 2). The result in SI Fig. 2d further supports the conclusion.

The authors should make revision in the main text to clarify the confusion. The paragraph that was unclear to me remain unchanged at all in the revised manuscript. I think the highlighted part should be briefly reflected in the related paragraph to improve the reasoning that lead to the conclusion.

Response: We thank the reviewer’s suggestion. We added the information in the revised manuscript. “The previous reports revealed that ADRP targets to adiposomes and that histones were found to localize to LDs via the anchor protein Jabba in *Drosophila*. Therefore, these results in Supplementary Figure 2e not only further suggest that the LD-targeting and DNA-binding domains of MLDS were at its N-terminus and C-terminus, respectively, which is consistent with the above results (Fig. 2f, i), but also...”

(3) We guess that because the fusion protein is instable. Full-length MLDS-N-H1 is degraded into several pieces. Based on the prediction of molecular weight, the highest band of the three bands is putative full-length MLDS-N-H1.

4. The authors concluded that the mRNA and protein levels of MLDS were significantly reduced in both the deletion and overexpression mutants, whereas in Fig 4f, it appears to me that KO and OE cases have less proteins in general, as evidenced by the bands on the top, especially as position of 95kDa and 55kDa. I think the authors should have a reference band for rigorous comparison.

Response: We thank the reviewer's critical comments. In fact, since bacterial LD study is still in early stage, we have not found any robust reference protein bands that can be used in bacterial LD fraction. Therefore, we think that comparing the total proteins of LD fraction in general is better way to use. In figure 4f, it is true that the proteins of KO lane were less than that of WT lane. But the decrease of MLDS was more significant and we repeated the experiment in SI F5d. The proteins of KO lane and WT lane were similar, and MLDS was also decreased dramatically. In figure 4f, the proteins of OE lane were similar with that of WT lane because of the large amount of MLDSR-GFP. Thus, protein levels of MLDS were significantly reduced in both the deletion and overexpression mutants.

I am not entirely convinced by comparing with the total protein level in each case. I suggest use some house-keeping protein as a normalization to quantity comparison, and put the quantification in the bar graph instead of having the readers judge by eyes.

Response: We thank the reviewer's critical comment. It's true that using some house-keeping proteins as normalization to quantity comparison is better. However, we have not found a house-keeping protein in bacterial LD fraction yet. We think that comparing with the total protein content in lanes is a suitable method. We agree with the reviewer's suggestion that "put the quantification in the bar graph instead of having the readers judge by eyes". Thus, we added the bar graph by quantifying the relative total protein content in lanes using Image J software in the necessary figures (Fig. 4f and SI fig. 5d) in the revised manuscript.

Minor:

Rate constant should be lower "*k*" instead of "K" in some of the figures.

Response: We thank the reviewer's suggestion. We corrected "K" to the "*k*" in Fig. 5d, e in the revised manuscript.

Reviewer #3 (Remarks to the Author):

This version is much improved, and I have no new critical comments. While I'm satisfied with most of my original issues, four of them remain and I ask the authors to again address them; otherwise, I'm fairly certain these point will confuse the readership. Numbers below correspond to my original critical comments. (I'm Reviewer 3.)

(2) figure 2d: a major gel shift occurs when the protein/DNA ratio reaches about 50. The authors term this (in their rebuttal letter) a "confused image." As this still indicates a major binding shift to me, the authors just need to address the point in the text or figure legend. If this gel behavior is an artifact at this ratio, this just needs to be stated as such.

Response: We thank the reviewer's suggestion and added the point in the legend of Fig. 2d in the revised manuscript. "Although these gel shifts were diffused when the protein/DNA ratio was from 20 to 40, these bindings could still be gradually increased since free monomer DNA was decreased with increased protein".

(6) The fact that the KO+MLDS doesn't fully rescue the KO phenotype just should be mentioned in the text, even if the explanation is not known. To me the omission is glaring.

Response: We thank the reviewer's suggestion and comment. We included "Re-expression of MLDS but not MLDS^N could partially rescue this phenotype (Fig. 3f and Supplementary Fig. 4a). The reason why MLDS could not fully recover the phenotype remains unknown" in the revised manuscript.

(7) figure 3g and 3j: Please just use images where the intensity of the main spot between WT and KO samples is similar, to allow the reader to directly compare the comet morphology.

Response: We thank the reviewer's suggestion. We replaced the images with the appropriate intensity images in the revised manuscript.

(10) In figure 5a, there are two 43 bp sequences at the bottom of the panel. It still is not clear to me which one of these is used in panel b (right). I didn't understand the explanation of the authors.

Response: We thank the reviewer's comment and understand the reviewer's doubt. In fact, they are two strands of one 43 bp sequence at the bottom of figure 5a. And the 103 bp is in the same case. We use dsDNA in EMSA in figure 5b. To avoid confusion to the reviewer and the readers, we added "5'" and "3'" in the terminus of the 103 bp and 43 bp sequences in figure 5a in the revised manuscript.

Reviewer #4 (Remarks to the Author):

The authors of the manuscript entitled "Lipid Droplets Mediate Genomic DNA Localization and Transcription to Promote DNA Stability and Bacterial Survival in

Extreme Environments” responded satisfactorily to the majority of my questions. I would have preferred if the authors had presented quantitative information gathered by techniques such as fluorometry, flow cytometry or real image analysis (as intensity ratios of 2 dyes present in the same image and not of measurements made with ImageJ) but I accept these results.

Response: We thank the reviewer’s useful suggestion and for accepting these results. We agree that using other methods and techniques can provide more evidence of quantification for TAG content and the level of DNA and LD interaction. We tried to analyze the TAG content (LipidTOX red stained) in RHA1 using flow cytometry before, but we failed to obtain reproducible results because the resolution of flow cytometry may not be enough to detect the fluorescence in bacteria. We will try these methods in our future study and thank the reviewer’s useful suggestion again.

However, I disagree with the explanation regarding electrostatic and hydrophobic interactions between the phospholipids monolayer of lipid droplets and DNA.

The authors claim that “LDs are coated by phospholipid monolayer membrane and phospholipids are negatively charged (phosphatidic acid, phosphatidyl inositol and phosphatidyl serine) or electrically neutral (phosphatidyl ethanolamine, phosphatidyl choline, and sphingomyelin). The DNA chain is negatively charged and thus there should not be electrostatic interaction between DNA and phospholipids membrane.”

Cellular membranes may contain zwitterionic phosphatidylcholines (PC) and phosphatidylethanolamine (PE) as well as anionic phosphatidylinositol (PI) and phosphatidylserine (PS). The phospholipid monolayer of the lipid droplets is complex, resulting from electrostatic and hydrophobic interactions between different phospholipids which have impact on lipid packing. The polyanionic DNA molecule interacts with the phospholipid headgroups and it’s repulsed by negative charges present in the phospholipids. However, the presence of counterions such as Ca ions may trigger electrostatic attraction (vide doi:10.1016/j.cis.2014.01.016; DOI: 10.1039/c3sm51419f; <http://dx.doi.org/10.1063/1.1325230>, doi: <http://dx.doi.org/10.1063/1.1436077>).

It is also known that lipid droplets in actinomycetes have a lipid core (with e.g. triacylglycerols and wax esters) covered by a monolayer of phospholipids, which prevents coalescence or denaturation of cytoplasmic proteins due to hydrophobic interactions (Hanish et al. 2006; doi: 10.1128/AEM.00584-06). Other studies have shown that most proteins that bind to the phospholipids monolayer of lipid droplets do it via hydrophobic interactions (e.g. Kory et al. 2016; <http://dx.doi.org/10.1016/j.tcb.2016.02.007>). Attractive interactions between the nitrogen bases of DNA and hydrophobic surfaces have been demonstrated by Cardenas et al. 2003 (DOI: 10.1021/la026747f).

The authors should read carefully the literature to assess the phospholipid composition of lipid droplets in *Rhodococcus jostii* RHA1 and discuss the electrostatic and hydrophobic interactions between those lipids and DNA and also between lipids and protein MLDS as its position depends on the same forces.

Response: We thank the reviewer's useful comment and suggestion. We added the information and re-organized the discussion in the revised manuscript. "Our current study presents that LDs recruit and protect genomic DNA through MLDS and the LD mimics adiposomes cannot bind DNA without MLDS or its substitute protein (Fig 2i, j and Supplementary Fig. 2d, e), which suggests that anchor proteins are necessary for DNA association to bacterial LDs. Furthermore, it has been reported that DNA was recruited to the bacterial cell membrane via the protein Noc and chromatin could interact with nuclear membrane by chromatin binding nuclear envelope proteins in eukaryotic cells, which further suggests that proteins are necessary for DNA interacting with phospholipid membrane. In addition, several previous reports revealed that the lateral distribution of anionic lipids within the membrane and membrane curvature could be affected in the presence of counterions like monovalent or divalent metal ions, polyamines, or cationic protein domains, and that DNA could contact to hydrophobic surfaces by cationic surfactants, which indicates that counterions and cationic surfactants may promote the interaction between DNA and phospholipid membrane. LDs are coated by phospholipid monolayer membrane and phospholipids are negatively charged (phosphatidic acid, phosphatidyl inositol and phosphatidyl serine) or electrically neutral (phosphatidyl ethanolamine, phosphatidyl choline, and sphingomyelin) under physiological conditions. Thus, the polyanionic DNA chain should not electrostatically interact with phospholipid membrane without proteins or other factors. Until now, however, it is not clear whether or what other proteins and/or factors are involved in mediating the interaction between DNA and LDs. We only found that MLDS can bridge the association between DNA and LDs in bacteria in the current study."

Thank you for your consideration of this revised manuscript.

REVIEWERS' COMMENTS:

Reviewer #2 (Remarks to the Author):

I am satisfied with the response from the authors and the revision in the manuscript. Only one more minor point: In the revised figure S2e, not clear to me what the color shades mean. Not the same shade is used for for each construct.

Reviewer #4 (Remarks to the Author):

The authors of the manuscript entitled "Lipid droplets mediate genomic DNA localization and transcription to promote DNA stability and bacterial survival", have answered my remarks in a satisfactory manner. I still think that the lipid character of LD must play an important role in the binding of DNA and other molecules, but I accept the results showing the role of the MLDS protein.

I think that the paper may be published. However, as I mentioned in my first review, the Hoechst® dye is blue. The authors should state in the manuscript that they changed the colour of the Hoechst® dye from blue to green by software and state how that was done, especially if changes in intensity were made.

Reviewer #2 (Remarks to the Author):

I am satisfied with the response from the authors and the revision in the manuscript. Only one more minor point: In the revised figure S2e, not clear to me what the color shades mean. Not the same shade is used for for each construct.

Response: We thank the reviewer's comment. In figure S2e, the different color shades just represent different lanes. To avoid the confusion, we change them and use the same shade in the revised manuscript.

Reviewer #4 (Remarks to the Author):

The authors of the manuscript entitled "Lipid droplets mediate genomic DNA localization and transcription to promote DNA stability and bacterial survival", have answered my remarks in a satisfactory manner. I still think that the lipid character of LD must play an important role in the binding of DNA and other molecules, but I accept the results showing the role of the MLDS protein.

I think that the paper may be published. However, as I mentioned in my first review, the Hoechst® dye is blue. The authors should state in the manuscript that they changed the colour of the Hoechst® dye from blue to green by software and state how that was done, especially if changes in intensity were made.

Response: We thank the reviewer's comments and for accepting the results. We agree that "the lipid character of LD must play an important role in the binding of DNA and other molecules" and we will research it in detail in the future study.

Furthermore, we added the information "For Hoechst staining of DNA, we replaced the original blue with the green in images by softWoRx 5.0 without any other changes to observe the association between the green of DNA and the red of LDs" in the Method of the revised manuscript.

Thank you for your consideration of this revised manuscript.